# Interview with an avatar: Comparing online and virtual reality perspective taking for gender bias in STEM hiring decisions

Cassandra L. Crone[1]*, Rachel W. Kallen[1,2]

1 Faculty of Medicine, Health and Human Sciences, School of Psychological Sciences, Macquarie University, Sydney, NSW, Australia, 2 Centre for Elite Performance, Expertise and Training, Macquarie University, Sydney, NSW, Australia

* cassandra-lynn.crone@students.mq.edu.au

**Data Availability Statement:** Data underlying the findings of the research has been made available via Mendeley Data. https://data.mendeley.com/

## Abstract

Virtual perspective taking can reduce unconscious bias and increase empathy and prosocial behavior toward individuals who are marginalized based on group stereotypes such as age, race, or socioeconomic status. However, the question remains whether this approach might reduce implicit gender bias, and the degree to which virtual immersion contributes to behavioral modulation following perspective taking tasks is unknown. Accordingly, we investigate the role of virtual perspective taking for binary gender using an online platform (Study 1) and immersive virtual reality (Study 2). Female and male undergraduates performed a simulated interview while virtually represented by an avatar that was either congruent or incongruent with their own gender. All participants rated a male and a female candidate on competence, hireability, likeability, empathy, and interpersonal closeness and then chose one of these two equivalently qualified candidates to hire for a laboratory assistant position in the male dominated industry of information technology. Online perspective taking did not reveal a significant influence of avatar gender on candidate ratings or candidate choice, whereas virtual reality perspective taking resulted in significant changes to participant behavior following exposure to a gender-incongruent avatar (e.g., male embodied as female), such that men showed preference for the female candidate and women showed preference for the male candidate. Although between-group differences in candidate ratings were subtle, rating trends were consistent with substantial differences in candidate choice, and this effect was greater for men. Compared to an online approach, virtual reality perspective taking appears to exert greater influence on acute behavioral modulation for gender bias due to its ability to fully immerse participants in the experience of (temporarily) becoming someone else, with empathy as a potential mechanism underlying this phenomenon.

## Introduction

Many individuals encounter recurrent experiences of prejudice and discrimination resulting from pervasive and harmful stereotypes about their membership in one or more groups that

datasets/626n8889yb/1 DOI: 10.17632/
626n8889yb.1.

**Funding:** The author(s) received no specific
funding for this work.

**Competing interests:** The authors have declared
that no competing interests exist.

are devalued in particular domains (e.g., education, government) or in society more broadly
[1]. Whether explicit or implicit, such biases serve to maintain inequities in the community
and workplace, disadvantaging individuals on the basis of their gender, race, age, disability,
sexuality, or social class [2]. These systemic inequities not only manifest as differential access
to freedoms, opportunities, benefits, and services, but are also reflected in the underrepresentation of such individuals within academic and professional domains [3, 4]. One such polarity is
maintained by *sexism*, encompassing prejudice, stereotyping, and discrimination based on
one's sex or gender identification. Sexism is further characterized by the belief that one gender,
typically male, is superior to other genders [5]. Moreover, treating maleness as normative
results in practices that disproportionately disadvantage women and girls as well as transgender, non-binary, and queer-gendered individuals [6]. In particular, science, technology, engineering, and math (STEM) fields continue to suffer from prevailing gender disparities on an
international scale, with less than 30% of professional STEM positions occupied by women [7,
8]. Moreover, interventions targeting workplace hiring and retention practices exhibit limited
success in resolving women's underrepresentation in STEM [9, 10].

In response to the limitations of current interventions, psychological research has begun to
investigate virtual perspective taking techniques aimed at modulating social biases. In online
virtual environments and in immersive virtual reality, *embodiment* refers to a perceptual illusion of ownership over a virtual body [11]. Concurrent with the experience of embodiment,
behavior is known to change to be consistent with digital representations of oneself [12].
When applied to race [13], age [14], and socioeconomic status [15], preliminary research has
demonstrated that visually altering an individual's identity during a virtual task, such that it is
incongruent with their own identity, may reduce implicit biases and increase empathy and
prosocial behavior. Although such virtual perspective taking may also be effective in modulating gender bias, limited research to date has examined how embodiment may be experienced
with regard to gender [16]. Moreover, no experimental studies, to our knowledge, have explicitly investigated the role of gender bias during face-to-face STEM hiring processes, in which
gender is difficult to conceal. Thus, to extend existing research on gender bias in STEM hiring,
the current investigation offers a comparison of virtual perspective taking for gender in an
online virtual environment and in fully immersive virtual reality by employing a simulated
STEM interview task.

## Gender bias and STEM hiring

The underrepresentation of women in STEM has been attributed to factors such as values [17],
lifestyle choices (e.g., motherhood, caretaking) [18, 19], culture (e.g., patriarchal social structures) [20, 21], occupational sorting [22, 23], social identity threat [24–26], and gender stereotypes [27–30]. However, certain research suggests that an overarching gender bias favoring
men, rather than explicitly disadvantaging women, may better account for these interrelated
and complex psychosocial factors [31]. Namely, women report reduced engagement and
greater expectation for discrimination following exposure to STEM gender bias [31–33].

The implications for decision making favoring men in STEM are widespread. Compared to
equally qualified men, women in STEM are less likely to receive funding for their research
[34], citations for their publications [35], occupy senior university and industry positions [36,
37], or be referred to by their professional titles [38]. When the same scientific abstract is
authored by a female, it is judged to be of poorer quality and less worthy of collaboration than
when authored by a male [39]. Female technology interns are rated by their superiors as having
lower aptitude than equivalent male interns [40], and young girls are given less scientific
instruction and rated as less capable of learning physics than identical young boys [41]. Taken

together these findings suggest that to better enable women's participation and progress in STEM, it is necessary to address the influence of gender-biased beliefs and attitudes within specific systemic processes.

Reducing and targeting gender bias evidenced in STEM professions continues to present an unresolved challenge. Currently, the gold-standard approach to combatting gender and other biases in the workplace is delivered by implicit bias training and education programs. These interventions aim to enhance one's awareness of their unconscious attitudes and beliefs about women in STEM and include exercises to alter these unconscious processes [42, 43]. Acute improvements in biased attitudes, diversity awareness, and hiring of individuals from marginalized groups have been exhibited among program participants [44, 45], yet changes to attitudinal and behavioral components do not persist in the long term [42]. Thus, the success of most diversity programs is, regrettably, transient, and repetition of training is sparse due to limited workplace resources (e.g., cost, time) [46]. Moreover, meta-analyses suggest that these programs do little to secure equitable hiring practices and benefit men more than women [9, 47–49].

In some cases, the visibility of diversity programs and equity policies within an organization can contribute to perceptions of fairness and safety among members of marginalized groups (e.g., women in STEM, people of colour) [50–52]. However, these same strategies can paradoxically compromise diversity and equity, as training and education programs can also activate gender stereotypes and normalise bias [53, 54]. For instance, the presence of organizational diversity policies led women to view sexist hiring as reasonable [50] and contributed to illusory perceptions of fairness among members of dominant groups (e.g., White men) [55]. Moreover, knowledge of diversity in hiring led dominant group members to exhibit concerns about unfair treatment (e.g., anti-White discrimination) and intolerance toward marginalized individuals, resulting in adverse hiring outcomes [55, 56]. Finally, STEM committees are known to promote fewer women for elite research positions when they perceive gender bias does not exist [57].

Highlighted in a meta-analysis of experimental research, STEM hiring procedures appear to be particularly vulnerable to gender bias favoring men [58]. This finding is supported by a series of hiring experiments in which gender was signalled by applicant names on otherwise identical application materials. Reviewers were more likely to select a male candidate over an identical female candidate for a university faculty position ($N = 238$ academic psychologists) [59], a laboratory manager position ($N = 127$ biology, physics, chemistry academics) [60], and a mentorship position ($N > 6500$ university academics) [61]. Moreover, male candidates were consistently rated as more competent and hireable than identical female candidates and given a higher starting salary [60, 62, 63]. Similar findings are demonstrated when participant gender is signalled by physical appearance. Despite equivalent performance by a female candidate on an arithmetic task, participants were more likely to hire a male candidate for a position in mathematics [64]. These experimental studies found no gender differences among participants, further suggesting that both men and women exhibit gender bias when faced with STEM hiring decisions.

## Virtual perspective taking, embodiment, and bias

Thus, we turn to virtual perspective taking as a potential means to enhance the participation and progression of STEM women. Previous work suggests that more immersive training may reduce gender bias in STEM [65–70]. Specifically, immersive perspective taking that employs behavioral rather than attitudinal or cognitive exercises can increase empathy, internal motivation against prejudice, and supportive, prosocial, or altruistic behaviors [71–75]. These effects

can persist up to eight months post-intervention and can also produce crossover effects, such that a behavioral perspective taking exercise (e.g., writing task, video, computer game) focused on one group can also increase positive outcomes toward members of other marginalized groups [73, 75].

Comparatively, traditional perspective taking exercises can result in acute reductions in implicit bias, prejudice, negative stereotypes, and perceived self-other differences, but these effects do not persevere after a four-day period [76, 77]. Given that traditional perspective taking exercises require individuals to imagine experiences of discrimination faced by marginalized individuals as well as their associated emotions, thoughts, and feelings [78], they may be less successful in the long term due to a lack of participant immersion and engagement. Remarkably, advances in virtual technologies can allow for increased immersion in perspective taking endeavors. By virtually representing individuals as the identity of another, they can more practically and viscerally experience what it might be like to encounter bias among everyday tasks within a simulated environment.

This technology has been employed in research investigating bias, prejudice, and discrimination across categories of age, race, gender, and socioeconomic status [79]. For example, participants embodied in a virtual avatar with an elderly face exhibited less negative stereotyping, more positive explicit attitudes, reduced implicit bias, and greater feelings of closeness toward elderly people after embodiment compared to individuals embodied in a young avatar or those who were not embodied [14, 80, 81]. Similarly, individuals embodied as a homeless person subsequently reported more positive attitudes and endorsed more supportive behaviors toward the homeless compared to those who were not embodied, with attitudinal change persisting up to eight weeks post-embodiment [15]. Following embodiment in a Black or dark-skinned avatar, White or light-skinned individuals also exhibit less implicit bias and more behavioral mimicry of other dark-skinned avatars, regardless of their actual race [13, 82, 83].

Collectively, the latter research suggests that virtual perspective taking shows promise for contributing to the mitigation of negative outcomes that emerge from unconscious biases. Nevertheless, the question remains whether such an approach can be applied to gender and its associated biases. To date, body ownership has been successfully induced for male participants in a virtual female body [16]. More recently, Lopez and colleagues [84] found that White men embodied as women during a virtual movement-based task exhibited an increase in implicit gender bias favoring men on the Gender-Career IAT compared to those embodied as men. Similarly, Schulze and colleagues [85] found that male ($N = 11$) and female ($N = 5$) participants indicated an increased preference for men in leadership positions following embodiment in any avatar within a virtual manager's office. However, these authors evaluated a limited number of participants spread across experimental groups and did not report whether IAT scores statistically differed pre- to post-embodiment. Conversely, Beltran and colleagues [86] found that participants who *directly* experienced a negative online workplace scenario reported reduced gender bias when represented by a female avatar compared to a male avatar.

Given the results of previous embodiment studies, these mixed preliminary findings for gender are somewhat unexpected. However, they highlight important considerations for applying a virtual perspective taking approach to the issue of gender bias in STEM. Schulze and colleagues [85] proposed that the use of a virtual manager's office may have induced a pre-existing implicit gender bias favoring men. Likewise, Groom and colleagues [87] found that participants exhibited increased racial bias against Black individuals after undertaking a mock interview task with a confederate. These simulated tasks represent everyday scenarios in which bias is prevalent, but few embodiment studies to date have included ecologically valid behavioral tasks as part of their investigations [15, 86–88]. Instead, embodiment studies rely on movement-based tasks to induce body ownership and on the IAT to measure implicit bias,

despite meta-analyses which find that the IAT remains a poor predictor of behavior [89–92]. Given that changes in implicit or explicit biases do not necessarily influence one another [93, 94], ecologically valid tasks and behavioral outcome measures may more accurately indicate bias during interpersonal interaction tasks undertaken in virtual environments.

## The relationship between embodiment, empathy, and behavioral change

The aforementioned embodiment studies additionally propose a novel consideration for the role of empathy in modulating bias and prosocial behavior. Research suggests that virtual perspective taking can affect empathy and prosocial behavior to a greater extent than traditional perspective taking exercises and that it is this phenomenon which might facilitate desired behavioral outcomes [15, 88, 95–97]. That is, virtual perspective taking may reduce bias directly through acute and sustained increases in empathy and indirectly through subsequent long-term changes in attitudes and pro-social behavior [15, 88]. In support of this assumption, neurobiological research has demonstrated that perspective taking and empathy recruit independent neural circuits, such that the cognitive processes involved in perspective taking can be differentiated from the affective processes that occur due to empathy [98]. Indeed, empathy can be learned or enhanced using embodiment in virtual reality by strategically training perspective taking [96]. While empathy following virtual embodiment is associated with reductions in negative bias related to social class [15], the influence of empathy as a causal mechanism in reducing gender and other biases remains to be seen.

## Overview of current studies

To investigate whether virtual perspective taking might reduce gender bias, we examine the role of embodiment for binary gender (i.e., female/male) in the context of the face-to-face STEM interview undertaken in an online virtual environment (Study 1) and in fully immersive virtual reality (Study 2). We aim to replicate and extend the methodologies previously employed in hiring studies [59–64] and contribute to the dearth of research comparing available delivery platforms for virtual perspective taking (e.g., computer-based, online, immersive virtual reality) [15, 88, 96]. Indeed, these studies indicate that implementing de-identified application materials alongside standardized evaluation criteria could reduce male preferential selection at the application stage, making selection for further assessment or interview more equitable. Thus, we begin to address the challenge of bias in the next critical stage of the hiring process, where in-vivo, person-to-person procedures dictate that gender and other physical characteristics are harder to conceal.

By employing a simulated interview task, we assess the effect of gendered embodiment, that is either *congruent* (i.e., male embodied as male) or *incongruent* (i.e., male embodied as female) with participant gender (self-identified), on the selection and rating of virtual STEM candidates. Across both experiments we anticipate that, compared to individuals embodied in a gender-congruent avatar, individuals embodied in a gender-incongruent avatar:

**H1:** Would be more likely to choose a female candidate over a male candidate.

**H2:** Would rate a female candidate as more competent, hireable, and likeable.

**H3:** These effects would be more pronounced for male participants.

**H4:** Participants would display more empathy and greater interpersonal closeness toward the candidate that is congruent with the gender of their assigned avatar.

Additionally, we explicitly examine the effects between participant gender, embodiment, and candidate ratings. Hiring studies suggest that women and men hold similar biases against

women in STEM [60, 63]. However, embodiment studies have yet to clarify whether virtual perspective taking exposures simply act to reduce bias, or more specifically, if individuals take on the identity of their assigned avatar and subsequently exhibit greater identification with co-actors of similar identities [13–15], Thus, it may be the case that for both women and men, embodiment reduces gender bias. Alternatively, embodiment may lead to identification with the gender of one's assigned avatar. To investigate these differential assumptions, we developed a competing hypothesis for H1 and H2 such that:

**H5:** Female participants will choose the male candidate over the female candidate and rate the female candidate less favorably when embodied in a male avatar, if they come

to identify with the male avatar to a greater extent.

## Study 1

To examine the effect of a virtual perspective taking task delivered online, we employed an interactive virtual environment suited to a two-dimensional display screen. Participants were virtually represented as a female or male avatar with first-person viewpoint and partial motor control/agency, which serve as the computer-based analogue for embodiment. In comparison to fully immersive virtual reality, online embodiment excludes certain features (e.g., real-time full-body motor control, three-dimensional viewpoint). Thus, we expected that the effects of online perspective taking may be less robust than virtual reality perspective taking due to a diminished perceptual sense of embodiment. Additionally, we assessed the proposition that virtual perspective taking tasks, such as those delivered using audio-visual components, may directly increase empathy, which then indirectly reduces bias and increases prosocial behavior [15, 66, 72, 95]. We expected that greater empathy and interpersonal closeness expressed toward a candidate would predict higher overall competence, hireability, and likeability ratings.

### Method

The procedures and ethical aspects of this study were approved by the Macquarie University Human Research and Ethics Committee (HREC). Reference Number: 52021609928600. All participants provided written voluntary informed consent prior to undertaking study procedures.

**Participants and design.** First-year undergraduate psychology students ($N = 65$; 41 female, 24 male; $M_{age} = 20.34$, $SD = 4.66$) were recruited from the university's online research participant database. In exchange for course credit, participants volunteered to take part in a study about how people behave when performing everyday social and professional activities in an online virtual environment and were asked to assist in conducting a simulated interview as part of a panel of two interviewers assessing two virtual candidates.

Participant gender was predetermined and self-selected as female or male. Participants were then randomly assigned to be either the female or the male interviewer, resulting in virtual embodiment that was either congruent (i.e., male with a male avatar, female with a female avatar) or incongruent (i.e., male with a female avatar, female with a male avatar) with the participant's gender. Therefore, this study employed a 2 (participant gender) x 2 (avatar congruency) between-subjects quasi-experimental design. Participant characteristics are summarized in Table 1, and there were no significant differences in participant demographics across experimental groups, suggesting that randomization was effective.

**Procedure.** Participants attended an online video conference with the researcher using Zoom, where they were provided with a link to the study's online virtual environment. Next,

**Table 1. Participant demographic characteristics by experimental condition in Study 1.**

| Measure | Male | | Female | | |
|---|---|---|---|---|---|
| | Congruent | Incongruent | Congruent | Incongruent | |
| | $n = 12$ | $n = 12$ | $n = 20$ | $n = 21$ | |
| | % or $M (SD)$ | | | | $\chi2$ or $F, p$ |
| Proportion right-handed | 100% | 91.7% | 100% | 95.2% | $3.12_{(4)}$, .538 |
| Race/Ethnicity | | | | | $25.57_{(16)}$, .060 |
| Asian | 8.3% | 25.0% | 10.0% | 19.0% | |
| White | 33.3% | 33.3% | 40.0% | 52.4% | |
| European | 16.7% | 25.0% | 30.0% | 14.3% | |
| Middle Eastern | 0% | 8.3% | 15% | 14.3% | |
| Other | 41.7% | 8.3% | 5% | 0% | |
| Age | 23.58 (8.71) | 18.83 (0.58) | 19.75 (3.40) | 19.90 (2.90) | $2.01_{(4, 64)}$, .104 |

participants were told that they would be collaborating in real time with three additional remotely located participants, that all participants would be represented by virtual avatars, and thus they would be randomly assigned to be either a female or a male interviewer on a two-interviewer panel. However, we used a planned deception and in truth, the second virtual interviewer and virtual candidates were pre-recorded and programmed into the virtual environment. Prior to undertaking the interview task, participants were required to read a job description outlining key criteria and skills required for a laboratory assistant job in programming and technology and to answer two engagement check questions to prime them to assess candidates for a position in information technology (IT). Upon entering the virtual environment, participants were instructed to undergo a 30-second exploration exercise of the virtual interview room and were guided through two practice interview questions. We specifically instructed them to look to their left to view their assigned avatar in a floor length virtual mirror. Participants were also told that the volume of their microphone would be tested during this exercise to enhance the belief that they would be communicating in real time with others located remotely. After completing these exercises, participants were verbally told by the experimenter that they would be connected with their peers and were shown a 10-second connection screen linking them to the second virtual interviewer and virtual candidates.

Participants undertook two separate interview sessions with each of the virtual candidates (one female, one male, presented randomly). During the interview task, participants delivered a set of four interview questions by alternating with the virtual interviewer, who was seated to their right and represented as the opposing binary gender to their assigned avatar. Participants' interview questions appeared on a tablet screen in the virtual environment, and they always asked the first and third questions. Participants were instructed to read their questions out loud and then listen to each of the candidate's responses without interrupting or asking any follow-up questions. Following completion of each candidate's interview, we asked participants to rate the candidate on measures assessing competence, hireability, and likeability and to report on empathy and interpersonal closeness with the candidates by making their selections on the questionnaires that were built into the virtual environment. After the second interview and candidate ratings were completed, participants were given the opportunity to review the job description before choosing which of the two candidates they would prefer to hire. Finally, participants provided information about their age, gender, and cultural background. All participants were debriefed as to the true nature of the study (i.e., to induce perspective taking) following the conclusion of these experimental procedures.

**Materials and apparatus.**   *Job description.* Modelled from an authentic job description at the university, this document outlined the key skills required for a research assistant position in human-machine systems, including experience in programming, data analyses, and research with human subjects. The job description was designed to prime participants to assess candidates for a position in information technology, as this area exhibits the largest disparities between women and men in STEM [7, 8]. To do this, we specified that a successful applicant would work in computer and cognitive science departments to develop artificial agents for human-machine interaction and virtual reality training systems, and job criteria were outlined by an academic who specializes in these areas of research. Three key skills were outlined: (i) research experience with human subjects, (ii) training in data and statistical analysis, (iii) experience with programming in virtual reality.

*Virtual environment.* The cross-platform gaming engine Unity-3D (version 2018.4 [LTS] Unity Technologies, San Francisco, USA) was used to create the virtual environment, designed to represent a modern professional faculty office and meeting workplace. The virtual environment featured a large desk, where candidates and interviewers conducted the interview while seated. To the left of the participant's avatar, a large mirror was featured on the wall to allow participants to view their virtual body and appearance. The environment was otherwise designed to be minimal and with neutral luminance so that it was not excessively interesting or distracting to participants when conducting the interviews. The virtual environment and experimental procedures are illustrated in Fig 1 and at the following video link: https://www.youtube.com/watch?v=FGfF_TIeajI.

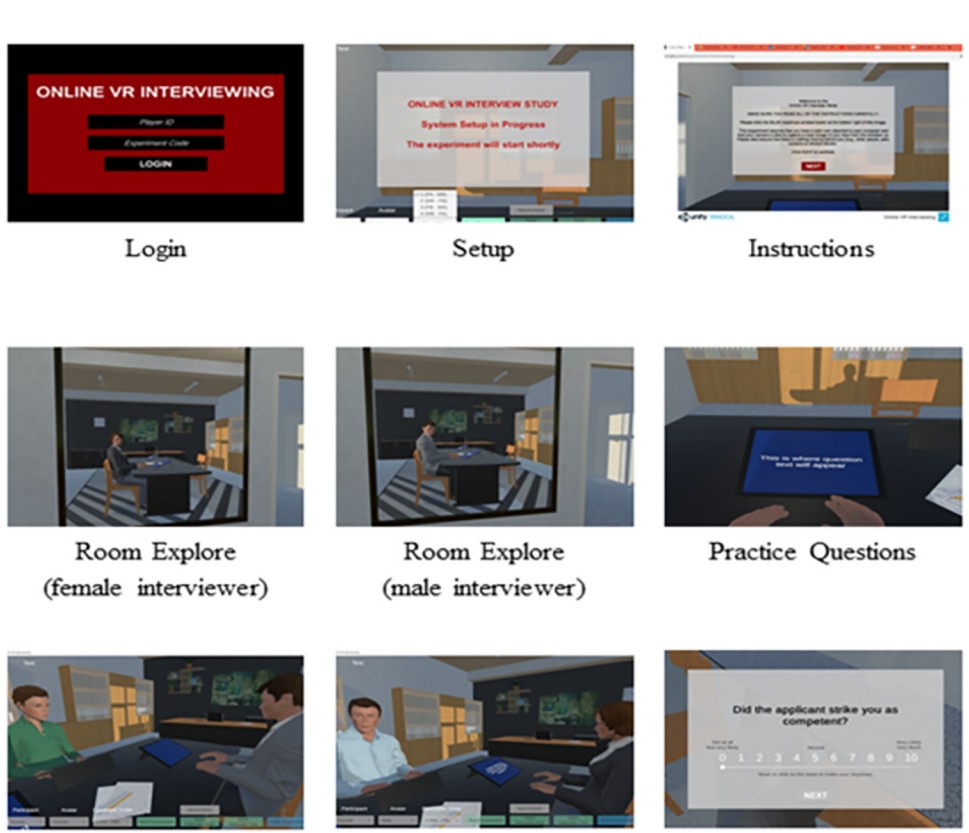

**Fig 1. Virtual scenes.** Visual still screen captures illustrating key study tasks undertaken in the virtual environment. Images are presented in sequence from top left to bottom right.

*Participant online platform.* A server-client application system was employed to conduct the study online. Participants viewed the virtual environment via a Unity Web-GL client application, which could be connected using the online link in any standard web-browser (e.g., Chrome, Firefox). Participants interacted with the environment using a mouse or trackpad to move their camera, navigate through instruction panels, and respond to candidate ratings.

*Experimenter desktop platform.* The experimenter, or server, version of the environment was compiled as a standalone desktop application. The experimenter would start this version of the environment before participants connected to the online application. Controls within the desktop application allowed the experimenter to set the participants' gender, assigned avatar, and order of candidate presentation as well as initiate pre-programmed avatar responses. This application also allowed the experimenter to view what the participant saw during the study (i.e., how the participant moved their camera view). The average latency between the participant (client) and experiment (server) applications was less than 100 ms, typically 30–60 ms, and operated at a fixed screen update frequency of 50 Hz.

*Interview questions.* A set of four interview questions were selected for use in this study. The first three questions were developed to target the key skills outlined in the job description (i.e., programming, data analysis, research with human subjects), and the final question was developed to target teamwork capabilities. These questions were modelled from a template of standardized questions used in interviews at the university where the research was conducted.

*Candidate responses.* Two sets of equivalent but distinct responses were used to represent recent graduates of a bachelor's degree in IT, including their education, work, and internship history. These responses constituted moderately, but not undeniably qualified candidates for the specified position. Candidate response set A and response set B were scripted to be equivalent in word count, timing, skill level, and qualifications. These candidate responses were then piloted for equivalency among a separate sample of university students and community members (N = 137), who were asked to rate each of the eight responses with reference to the job description from 1 (fails to meet requirements) to 4 (superior) and then choose which of the two fictional graduates they would prefer to hire. Paired-samples t-tests were conducted on the average ratings for each response set, with the final version demonstrating equivalency between response set A (M = 11.93, SD = 2.54) and response set B (M = 12.19, SD = 2.34, $t_{(42)}$ = -0.86, p = .392). Response sets were counterbalanced in combination with candidate gender (female, male) to control for primacy, recency, and comparative effects.

*Avatars.* Adobe Fuse software and Mixamo (https://www.mixamo.com) were used to create avatar models and included a humanoid rig (bone structured) that allowed for full-body, head, and facial animation. Avatars were designed to meet average objective attractiveness standards to ensure that participants would attend to the candidate's gender and qualifications rather than a high or low degree of attractiveness. Avatars were also designed to be as neutral as possible (e.g., skin tone, eye color, hairstyle, makeup, clothing color/pattern), but were dressed in a business casual, professional manner (Fig 2). We piloted a selection of female and male avatars among a separate sample of community members (N = 53), who rated eight avatars from 1 (very unattractive) to 10 (very attractive). Paired t-tests with a Bonferroni correction (α = .003) were conducted on these responses to choose two female and two male avatars with equivalent attractiveness ratings to represent the female (M = 6.46, SD = 1.75, $t_{(25)}$ = 0.45, p = .658) and the male interviewers (M = 6.27, SD = 1.59) and the female (M = 6.58, SD = 1.42, $t_{(25)}$ = 2.14, p = .042) and the male (M = 5.58, SD = 1.96) candidates.

Participants viewed the female and male candidates only during their respective interviews. After being randomly assigned to be either the female or the male interviewer, participants could view the virtual co-interviewer seated to their right during the interviews. The co-interviewer was always the opposing binary gender to the participant's assigned avatar. While in

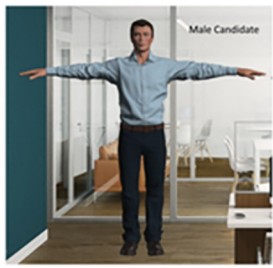
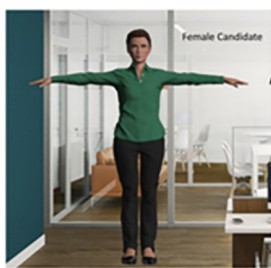
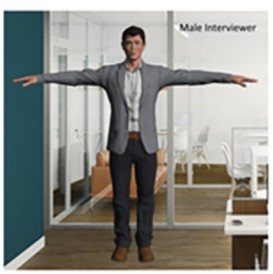
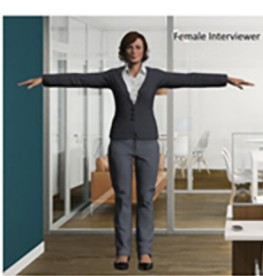

**Fig 2. Female and male avatars.**

the virtual environment, participants viewed their assigned avatar from a first-person perspective and could additionally view their avatar's full body at any time by looking into the mirror on the left.

*Animation, audio, and programming.* The virtual co-interviewer and virtual candidates were programmed to speak after a randomized 1–3 second interval, initially cued by the experimenter. Consistent with previous human-AI and human-avatar interaction studies [99, 100], facial animations and eye-looking gestures were programmed using SALSA Lip-Sync software (https://crazyminnowstudio.com/ unity-3d/lip-sync-salsa/). This software uses machine learning methods to simulate realistic facial animations from audio recordings, which were provided by young adult Australian volunteers (two female, two male). Volunteers used a neutral but professional tone of voice as if they were attending an interview. These audio recordings were integrated and synchronized with facial animations in the Unity platform to create the avatar speech cues. To further replicate realistic eye-looking behavior, the virtual candidates were programmed to look at one interviewer when being asked a question and then to intermittently shift their gaze between both interviewers when giving a response.

**Measures.** *Candidate evaluation ratings.* Candidates were rated on their perceived competence (i.e., skill, capability), hireability (i.e., job fit, suitability), and likeability (i.e., personality, professional manner) on an 11-point Likert-type scale from 0 (*not at all/not at all likely*) to 10 (*very much/very likely*). Each of these scales is comprised of three items previously validated in a series of hiring studies [101–103]. Average scores for each scale were calculated from participant responses to each of the scale's three items, with higher scores indicating greater perceived qualification for the job described. Items were modified for the present study to suit the specified job position and the target virtual candidates who represented recent graduates of a bachelor's degree. These scales have previously demonstrated good to excellent internal consistency (competence: $\alpha$ = .92-.93, hireability: $\alpha$ = 91-.94, likeability: $\alpha$ = 87-.93) [60, 63].

*Empathy.* Three items from Batson and colleagues [104] were used to assess the degree of felt empathy toward each of the candidates on an 11-point Likert-type scale from 0 (not at all) to 10 (very much). Participants indicated the degree to which they felt compassionate, warm, and sympathetic toward the candidate, with higher average scores indicating higher empathy.

Items were modified for the present study to suit the target applicants. This measure has previously demonstrated adequate internal consistency for both the full form ($\alpha$ = .85-.94) [104, 105] and a modified three-item short form ($\alpha$ = .85–88) [66].

*Interpersonal closeness.* The Inclusion of Other in the Self scale (IOS) [106], a 1-item pictorial measure of subjective interpersonal closeness, was used to assess relationship interconnectedness between the participant and each of the candidates. Participants selected the illustrated degree of overlap between two circles that best represented each of these relationships on a scale from 1 (no overlap) to 7 (complete overlap). This measure has previously demonstrated alternate-form reliability, test-retest reliability, and convergent, discriminant, and predictive validities [106, 107].

## Results

**Preliminary analyses.** A series of paired t-tests were conducted to check for any primacy and recency effects. Results indicated no significant primacy or recency effects for order of candidate presentation for competence ($t_{(64)}$ = 0.96, $p$ = .340), hireability ($t_{(64)}$ = -0.10, $p$ = .918), likeability ($t_{(64)}$ = 1.18, $p$ = .242), empathy ($t_{(64)}$ = -0.17, $p$ = .867), or interpersonal closeness ($t_{(64)}$ = -1.42, $p$ = .160). Additionally, there were no significant differences in ratings between response set A and response set B for competence ($t_{(64)}$ = -0.71, $p$ = .481), hireability ($t_{(64)}$ = 0.03, $p$ = .978), likeability ($t_{(64)}$ = -1.01, $p$ = .317), empathy ($t_{(64)}$ = -0.17, $p$ = .867), or interpersonal closeness ($t_{(64)}$ = 0.54, $p$ = .592).

A 2 x 2 Pearson's chi-squared ($\chi2$) test was used to assess whether participants exhibited preference for either response set when the candidate was female and when the candidate was male. Results suggested that there were no substantial preferences for either response set. When the female candidate was set A (and the male candidate was set B), participants chose each response set a similar number of times (A = 16, B = 16), whereas when the female candidate was set B (and the male candidate was set A), participants chose set B more often (A = 11, B = 22). However, these differences were not significant, $\chi2$ (1, N = 65) = 1.86, $p$ = .173. These results are consistent with pilot testing and suggest that counterbalancing was effective.

**Candidate choice.** To test the hypothesis that participant gender would interact with congruency to influence candidate choice, we conducted a 2 x 2 logistic regression. Table 2 summarises the frequency of candidate choice by participant gender in the congruent and incongruent embodiment conditions. Although the interaction term was not significant, the observed pattern of results suggested that participants in the incongruent condition had greater odds of choosing the female candidate compared to participants in the congruent condition ($b$ = 0.09, $SE$ = 0.69, $p$ = .890, OR = 1.09), but this effect was reduced for male compared to female participants ($b$ = -0.44, $SE$ = 0.68, $p$ = .674, OR = 0.64), such that men were more

**Table 2. Candidate choice by embodiment and participant gender in Study 1.**

| | Candidate Choice | | |
|---|---|---|---|
| | **Male** | **Female** | **$\chi2$, $p$** |
| Congruent | | | 0.42, .515 |
| Male | 4 | 8 | |
| Female | 9 | 11 | |
| Incongruent | | | 0.01, .947 |
| Male | 5 | 7 | |
| Female | 9 | 12 | |
| Total | 27 | 38 | |

**Table 3. Summary of candidate ratings by experimental group in Study 1.**

| Measure | Female Participants | | | | Male Participants | | | |
|---|---|---|---|---|---|---|---|---|
| | Congruent | | Incongruent | | Congruent | | Incongruent | |
| | Male | Female | Male | Female | Male | Female | Male | Female |
| | *M (SD)* | | | | | | | |
| Competence | 8.57 (1.11) | 8.83 (0.86) | 8.78 (1.02) | 8.75 (0.77) | 8.67 (0.57) | 8.78 (0.95) | 8.81 (0.82) | 8.50 (1.30) |
| Hireability | 8.57 (0.93) | 8.60 (1.16) | 8.32 (1.15) | 8.40 (1.01) | 7.92 (1.05) | 8.25 (1.46) | 8.08 (1.27) | 8.31 (1.11) |
| Likeability | 7.40 (1.26) | 7.23 (1.70) | 7.38 (1.51) | 7.14 (1.50) | 6.44 (1.77) | 6.50 (2.00) | 7.08 (1.67) | 7.61 (1.20) |
| Empathy | 6.40 (1.53) | 6.02 (2.03) | 5.94 (1.98) | 5.81 (1.56) | 5.14 (2.10) | 4.47 (2.09) | 5.22 (1.26) | 5.69 (1.16) |
| IOS | 3.10 (1.29) | 2.60 (1.23) | 2.62 (1.43) | 2.48 (1.33) | 2.42 (1.00) | 2.42 (1.38) | 3.17 (1.03) | 3.00 (1.28) |

likely to choose the male candidate. That is, women were 22% more likely and men were 100% more likely to choose the female candidate in the congruent condition, whereas men were only 40% more likely to do so in the incongruent condition compared to women at 33%. However, this model ($\chi 2_{(3)} = 0.45$, $p = .929$) and its individual predictors were not significant. With 0.7, 0.8, and 0.9 corresponding to acceptable, excellent, and outstanding strength of discrimination, respectively [108], model fit was poor with an area under the curve (AUC) of .54.

**Candidate ratings.** To test the hypotheses regarding candidate ratings for competence, hireability, likeability, empathy, and interpersonal closeness, we conducted a series of three-way mixed ANOVAs. As the composite rating scores for competence and hireability exhibited substantial negative skew, these scores were transformed using a log transformation with a reflection to better conform to the normality requirement. For each outcome variable, a 2 x 2 x 2 model was specified with participant gender and congruency as the two between-subjects factors and composite rating scores for the female and male candidates as the within-subjects factor.

Average candidate ratings by participant gender and embodiment are summarized in Table 3, and results of mixed ANOVA analyses are presented in Table 4. For models assessing competence (Fig 3A), hireability (Fig 3B), likeability (Fig 3C), and interpersonal closeness (Fig 3E), no significant differences were demonstrated between subjects for gender or congruency

**Table 4. Mixed ANOVA models for candidate ratings following online perspective taking.**

| Measure | Competence | Hireability | Likeability | Empathy | IOS |
|---|---|---|---|---|---|
| | $F_{(1, 61)}$ | $F_{(1, 61)}$ | $F_{(1, 61)}$ | $F_{(1, 61)}$ | $F_{(1, 61)}$ |
| Gender | 0.05 ($p = .825$, $\eta_p^2 < .01$) | 1.67 ($p = .201$, $\eta_p^2 = .03$) | 1.14 ($p = .289$, $\eta_p^2 = .02$) | 4.84 ($^*p = .032$, $\eta_p^2 = .07$) | 0.03 ($p = .862$, $\eta_p^2 < .01$) |
| Embodiment | 0.07 ($p = .800$, $\eta_p^2 < .01$) | 0.15 ($p = .697$, $\eta_p^2 < .01$) | 1.33 ($p = .253$, $\eta_p^2 = .02$) | 0.15 ($p = .702$, $\eta_p^2 < .01$) | 0.39 ($p = .537$, $\eta_p^2 = .01$) |
| Candidate | 0.18 ($p = .670$, $\eta_p^2 < .01$) | 2.12 ($p = .150$, $\eta_p^2 = .03$) | 0.05 ($p = .817$, $\eta_p^2 < .01$) | 0.90 ($p = .348$, $\eta_p^2 = .01$) | 1.91 ($p = .172$, $\eta_p^2 = .03$) |
| Gender*Embodiment | 0.01 ($p = .989$, $\eta_p^2 < .01$) | 0.33 ($p = .568$, $\eta_p^2 = .01$) | 1.71 ($p = .195$, $\eta_p^2 = .03$) | 1.43 ($p = .236$, $\eta_p^2 = .02$) | 2.73 ($p = .103$, $\eta_p^2 = .04$) |
| Candidate*Gender | < 0.01 ($p = .989$, $\eta_p^2 < .01$) | 0.94 ($p = .336$, $\eta_p^2 = .02$) | 1.65 ($p = .204$, $\eta_p^2 = .03$) | 0.18 ($p = .673$, $\eta_p^2 < .01$) | 0.66 ($p = .420$, $\eta_p^2 = .01$) |
| Candidate*Embodiment | 1.78 ($p = .187$, $\eta_p^2 = .03$) | 0.50 ($p = .483$, $\eta_p^2 = .01$) | 0.27 ($p = .604$, $\eta_p^2 < .01$) | 3.51 ($p = .066$, $\eta_p^2 = .05$) | 0.11 ($p = .746$, $\eta_p^2 < .01$) |
| Candidate*Gender*Embodiment | 0.03 ($p = .874$, $\eta_p^2 < .01$) | 0.15 ($p = .702$, $\eta_p^2 < .01$) | 0.50 ($p = .482$, $\eta_p^2 = .01$) | 1.40 ($p = .241$, $\eta_p^2 = .02$) | 0.80 ($p = .375$, $\eta_p^2 = .01$) |

*Note*. Between-subjects factors: gender (female, male), embodiment (congruent, incongruent); within-subjects factor: candidate (female, male)

$^*p < .05$

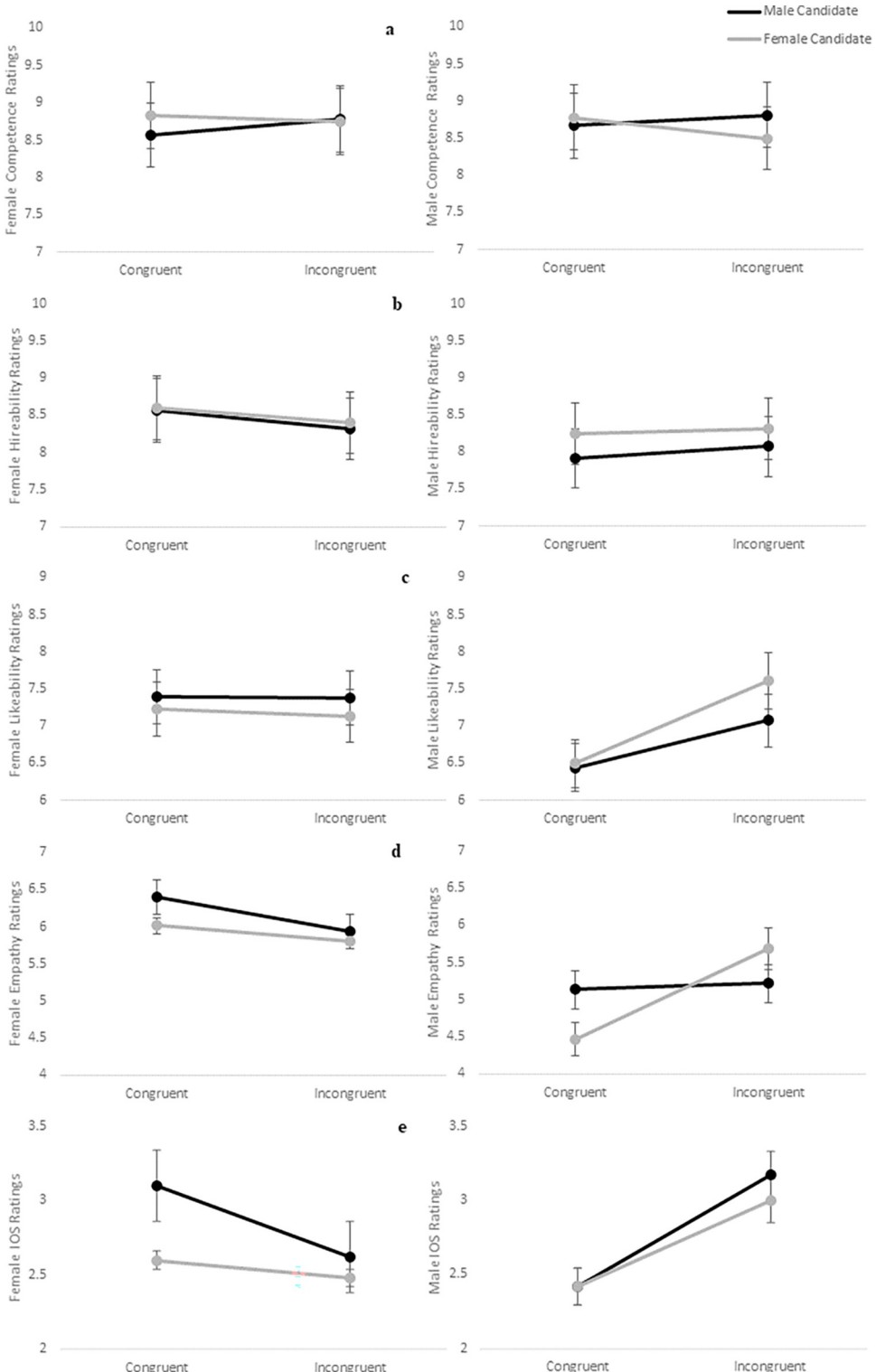

**Fig 3. Candidate ratings following online perspective taking.** Comparison of female (left) and male (right) participant ratings by embodiment for the virtual candidates as illustrated for a) competence; b) hireability; c) likeability; d) empathy; e) interpersonal closeness.

**Table 5. Multiple regression models for the female candidate.**

| Dependent Variable | Predictors | R2% | b | SE | B | $t_{(df)}$ | p | 95%CI | |
|---|---|---|---|---|---|---|---|---|---|
| | | | | | | | | Lower | Upper |
| Competence | | 1.7 | | | | | | | |
| | Empathy | | 0.07 | 0.07 | 0.14 | 0.98 | .331 | -0.07 | 0.21 |
| | IOS | | -0.02 | 0.10 | -0.02 | -0.14 | .886 | -0.22 | 0.19 |
| Hireability | | 10.8 | | | | | | | |
| | Empathy | | 0.23 | 0.09 | 0.36 | 2.69 | .009* | 0.06 | 0.40 |
| | IOS | | -0.08 | 0.12 | -0.09 | -0.69 | .490 | -0.32 | 0.16 |
| Likeability | | 52.8 | | | | | | | |
| | Empathy | | 0.56 | 0.09 | 0.63 | 6.48 | .000** | 0.39 | 0.74 |
| | IOS | | 0.23 | 0.12 | 0.18 | 1.84 | .070 | -0.02 | 0.47 |

*Note.*

$^*p < .05$

$^{**}p < .001.$

nor within-subjects among rating scores for the female and male candidate. For empathy (Fig 3D), a significant main effect was found for gender, such that women rated the candidates higher on empathy than did men overall, and the two-way interaction between embodiment and candidate ratings, which approached but did not reach the nominated statistical significance ($\alpha = .05$), suggested that embodiment might also exert some influence on empathy ratings.

**Associations between empathy and interpersonal closeness with candidate ratings.** A series of multiple regression analyses were conducted to assess whether empathy and interpersonal closeness uniquely predicted competence, hireability, and likeability ratings for the female and the male candidate. Continuous variables were mean centred, as interpersonal closeness was measured on a 7-point scale and did not include a value of zero, while the remaining factors were measured on an 11-point scale and included a value of zero. Bootstrapping with 1000 resamples was applied to models assessing competence and hireability of the male candidate, as residuals for these models were not normally distributed. Greater empathy, but not greater overlap with the candidates, predicted higher ratings for competence, hireability, and likeability of the male candidate and higher ratings for hireability and likeability, but not competence of the female candidate, summarized in Tables 5 and 6.

**Discussion.** Our assessment of candidate choice suggested that neither congruency nor participant gender had a substantial influence on preference for the female or male candidate following virtual perspective taking in an online environment. Consistent with an in-group bias for one's own gender (i.e., females prefer other females, males prefer other males) [109], men remained more likely to choose the male candidate compared to women, who were more likely to choose the female candidate. Overall, participants chose the female candidate more often than the male candidate, potentially suggesting socially desirable responding.

Between-group trends in candidate ratings suggested that women and men may be influenced differently by online virtual perspective taking for gender. Women tended to rate both candidates similarly, regardless of whether they experienced congruent or incongruent embodiment. Likewise, men tended to rate the male candidate similarly in both conditions, but following incongruent embodiment, they rated the female candidate higher on all measures except competence, which decreased. Consistent with previous research [110], women reported significantly more empathy toward the candidates than did men on average. However, the two-way interaction ($p = .066$) that approached but did not reach the nominated

**Table 6. Multiple regression models for the male candidate.**

| Dependent Variable | Predictors | R2% | b | SE | B | t(df) | p | 95%CI | |
|---|---|---|---|---|---|---|---|---|---|
| | | | | | | | | Lower | Upper |
| Competence | | 7.9 | | | | | | | |
| | Empathy | | 0.17 | 0.07 | 0.32 | 2.13 | .032* | 0.04 | 0.33 |
| | IOS | | -0.06 | 0.11 | -0.08 | -0.53 | .555 | -0.29 | 0.14 |
| Hireability | | 14.8 | | | | | | | |
| | Empathy | | 0.26 | 0.09 | 0.43 | 2.95 | .007* | 0.08 | 0.44 |
| | IOS | | -0.07 | 0.11 | -0.08 | -0.56 | .516 | -0.28 | 0.13 |
| Likeability | | 61.3 | | | | | | | |
| | Empathy | | 0.61 | 0.08 | 0.71 | 7.33 | .000** | 0.44 | 0.77 |
| | IOS | | 0.13 | 0.12 | 0.11 | 1.14 | .260 | -0.10 | 0.37 |

*Note.* Bootstrapped 95% confidence intervals are reported for competence and hireability models.

*p < .05

**p < .001.

statistical significance highlighted that participants reported more empathy toward the female candidate following incongruent embodiment, but more empathy toward the male candidate following congruent embodiment. An examination of the three-way trends suggested that this effect was driven by the responses of the male, but not female participants. Taken together, these trends suggest that online virtual perspective taking could have some effect on changing men's perceptions of a female candidate, but little effect on women's perceptions of a candidate based on gender presentation.

Finally, regression analyses revealed that greater empathy, but not interpersonal closeness felt toward the candidates predicted higher ratings for competence, hireability, and likeability of the male candidate and higher ratings for hireability and likeability, but not competence of the female candidate. Empathy uniquely contributed to a large proportion of the variance in candidate likeability (52.8–61.3%) but a smaller proportion of the variance in candidate competence and hireability (1.7%-14.8%). Consistent with van Loon and colleagues [88], online virtual perspective taking may sufficiently increase empathy, but like previous research [60, 63], this may not transfer to more equitable perceptions of female and male competence for a STEM position.

## Study 2

Study 2 replicates Study 1 in fully immersive virtual reality. Again, we test our hypotheses regarding candidate choice, candidate ratings (i.e., competence, hireability, likeability, empathy, interpersonal closeness), and two- and three-way interactions. Since the HMD virtual experience generates immersive components unavailable in an online environment (e.g., synchronized full-body motion control, mobile three-dimensional environment exploration, first-person co-location), we could better evaluate the proposed relationship between virtual perspective taking, empathy, and behavioral change in Study 2. Previous research using fully immersive virtual reality suggests that the experience of altered self-perception during and after embodiment in virtual reality leads to the positive behavioral outcomes (e.g., reduced bias) we observe toward specific targets (e.g., marginalized identities) *via* increases in empathy [15, 88, 95–97]. Given results obtained in Study 1, we expected that empathy, but not interpersonal closeness, would mediate the relationship between virtual immersion and candidate ratings.

## Method

All participants provided written voluntary informed consent prior to undertaking these study procedures and received course credit in exchange for their participation. There was no overlap between those who participated in Study 1 and Study 2, and experimental procedures, materials, and measures remained as outlined in Study 1 with the following variations.

**Participants and design.**   Noting the limited sample size of Study 1, we aimed to recruit 120 participants (60 female, 60 male) to provide adequate statistical power across the four experimental groups (i.e., female congruent, female incongruent, male congruent, male incongruent). To minimise participant risk, individuals with a history of motion sickness in virtual reality or a history of neurological, neuromuscular, or musculoskeletal disorders were precluded from participating in this study, and as such, these criteria were listed as eligibility requirements.

As we were primarily interested in comparing differences in the effect of embodiment between women and men, five participants who indicated that their gender was non-binary or gender fluid were excluded from the analyses. The final sample ($N = 131$) comprised 67 female and 64 male participants ($M_{age} = 20.92$, $SD = 5.09$). Participant characteristics are summarized in Table 7. There were no significant differences in participant demographics across experimental groups, suggesting randomization was effective.

**Procedure.**   First, participants read the job description outside of virtual reality. Participants were then seated at a table and entered the virtual environment using the HTC Vive Pro HMD. In a virtual waiting room, participants undertook eye-tracking calibration exercises and performed a standardized 60-second movement integration exercise while viewing their assigned avatar in the mirror in front of them. After proceeding to the interview room, participants undertook the 30-second room exploration, practiced assigned interview questions, interviewed the two virtual candidates, and completed candidate rating questionnaires (i.e., competence, hireability, likeability, empathy, interpersonal closeness). After exiting virtual reality, participants reviewed the job description and chose which of the two candidates they

**Table 7. Participant demographic characteristics by experimental condition in Study 2.**

| Measure | Male | | Female | | |
|---|---|---|---|---|---|
| | Congruent | Incongruent | Congruent | Incongruent | |
| | $n = 31$ | $n = 33$ | $n = 34$ | $n = 33$ | |
| | % or $M$ (SD) | | | | $\chi2$ or $F$, $p$ |
| Proportion right-handed | 90.3% | 90.9% | 91.2% | 87.9% | $0.24_{(3)}$, .970 |
| Race/Ethnicity | | | | | $3.47_{(6)}$, .748 |
| Asian | 25.8% | 27.3% | 26.5% | 42.4% | |
| White | 41.8% | 42.4% | 38.2% | 27.3% | |
| Other | 32.3% | 30.3% | 35.3% | 30.3% | |
| Age | 21.38 (6.92) | 22.33 (5.19) | 20.85 (4.77) | 19.15 (2.08) | $2.34_{(3, 127)}$, .077 |
| VR Experience | | | | | |
| Proportion yes | 58.1% | 66.7% | 38.2% | 51.5% | $6.01_{(3)}$, .111 |
| Proficiency | 1.26 (0.56) | 1.18 (0.40) | 1.08 (0.28) | 1.12 (0.49) | $0.54_{(3, 67)}$, .660 |
| Embodiment | 4.63 (0.66) | 4.59 (0.55) | 4.67 (0.67) | 4.82 (0.76) | $0.75_{(3, 127)}$, .525 |
| Body Ownership | 4.24 (0.93) | 4.15 (0.82) | 4.06 (1.23) | 4.51 (1.02) | $1.23_{(3, 127)}$, .301 |
| Agency/Control | 5.11 (0.64) | 5.14 (0.71) | 5.39 (0.70) | 5.19 (0.80) | $1.04_{(3, 127)}$, .376 |

*Note*. Other comprises Aboriginal ($n = 3$), European ($n = 17$), Hispanic ($n = 3$), Middle Eastern ($n = 14$), and mixed ($n = 5$) family/cultural backgrounds. VR = virtual reality.

would prefer to hire. Finally, participants completed a subjective measure of embodiment and indicated their age, gender, family/cultural background, left- or right-hand dominance, and previous experience with virtual reality, including their level of experience from 1 (*beginner*) to 3 (*advanced*).

**Materials and apparatus.** *Head-mounted display*. The HTC Vive Pro Eye virtual reality system is comprised of a headset including attachable headphones for audio output, two hand controllers, and two motion-tracking base stations. While seated and holding onto the hand controllers, participants interacted with the virtual environment and other virtual components viewed through the front-facing camera of the headset (110-degree visual field, 1440 x 1600 resolution per eye, 90 Hz refresh rate). The trigger on the right-hand controller was used to make selections and navigate through the environment during tasks. Base stations were mounted laterally at the top of the left- and right-side walls to track the spatial position of the headset and the hand controllers, allowing for real-time control of participant avatar motion (e.g., head and hand movements).

*Virtual environment*. Unity-3D (version 2019.4 [LTS], Unity Technologies, San Francisco, USA) was used to host the virtual environment and all integrated programming components (e.g., audio recording, avatars, animations, experimenter controls, participant instructions, questionnaires, experimental tasks) on the desktop computer in the lab. A waiting room scene (Fig 4) was added to Study 2 to conduct eye-tracking and motion integration exercises before entering the main interview room scene. Participants undertook these exercises while facing a full-length virtual mirror where they could view their avatar moving with them in real time.

*Movement integration task*. Consistent with previous virtual embodiment studies [13, 81–83, 87, 88], participants were instructed to complete a standardized movement integration task upon first viewing their avatar in the waiting room mirror to enhance body ownership and virtual immersion. As participants were seated throughout the study, they were required to move only their head and arms. Participants were instructed to follow the experimenter's verbal commands for the 60-second exercise, during which they were told to look left, right, up, and down and to move their left and right arms independently and then simultaneously in lateral and horizontal planes.

**Measures.** *Embodiment*. The degree of virtual immersion was measured on a 7-point Likert scale from 1 (*strongly disagree*) to 7 (*strongly agree*) using nine items from Gonzalez-Franco and Peck's [111] virtual reality embodiment questionnaire. Participants selected their level of agreement with each statement regarding their virtual body during the experimental tasks. Five items target body ownership, and four items target agency and motor control. Although validity and reliability of Gonzalez-Franco and Peck's [111] originally proposed items have been numerically investigated using principal components and confirmatory factor analyses [112, 113], no research to date has sufficiently assessed the psychometric properties of this measure.

## Results

**Preliminary analyses.** As in Study 1, paired samples t-tests revealed no significant differences in candidate ratings for the candidate presented first compared to the candidate presented second for competence ($t_{(130)}$ = 2.27, $p$ = .024), hireability ($t_{(130)}$ = 2.03, $p$ = .045), and empathy ($t_{(130)}$ = -0.45, $p$ = .652). However, there was a significant order effect for likeability ($t_{(130)}$ = 3.32, $p$ = .001) and for interpersonal closeness ($t_{(117)}$ = -3.08, $p$ = .003), using the Bonferroni corrected alpha ($\alpha$ = .005), which suggested that the candidate presented first was rated significantly higher on likeability but significantly lower on the IOS compared to the candidate presented second, regardless of candidate gender or response set. Thus, order of candidate

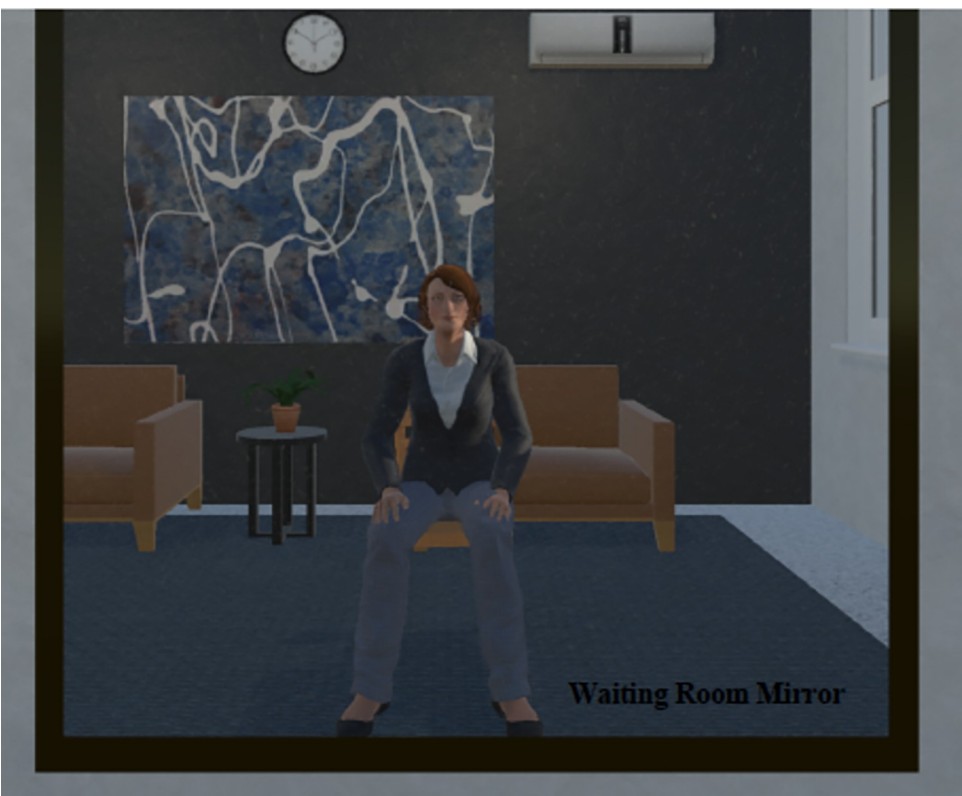

**Fig 4. Virtual waiting room scene.**

presentation was entered as a covariate in the mixed ANOVA models assessing ratings for likeability and interpersonal closeness. Additionally, due to a technical error in data recording, 15 participants did not provide IOS ratings for the second candidate, and as such, these cases were excluded from analyses using the IOS.

Consistent with pilot testing, there were no significant differences in candidate ratings for response set A compared to response set B for competence ($t_{(130)}$ = -0.67, $p$ = .502), hireability ($t_{(130)}$ = -0.96, $p$ = .338), likeability ($t_{(130)}$ = -1.02, $p$ = .310), empathy ($t_{(130)}$ = -0.59, $p$ = .555), and interpersonal closeness ($t_{(117)}$ = -0.09, $p$ = .929). Furthermore, the Pearson's $\chi2$ test revealed that there were no significant preferences for response set A or response set B $\chi2$ (1, $N$ = 131) = 0.39, $p$ = .535, such that participants chose each response at a similar frequency when the female was A and the male was B (A = 37, B = 32) compared to when the female was B and the male was A (A = 28, B = 34).

**Candidate choice.** Table 8 summarises the frequency of candidate choice by participant gender in the congruent and incongruent embodiment conditions. Binary logistic regression analyses revealed that although the interaction between participant gender and embodiment was not significant ($b$ = -1.22, $SE$ = 0.73, $p$ = .094, $OR$ = 0.30), the interaction exerted a suppressing effect and thus, was retained in the final model to accurately summarise the data. The full interaction model was significant ($\chi2_{(3)}$ = 7.94, $p$ = .047) and provided better model fit than the main effects only model ($\chi2_{(3)}$ = 5.09, $p$ = .078), with an AUC of .63. There was a significant main effect of gender, such that men were more likely to choose the female candidate on average ($b$ = 1.41, $SE$ = 0.53, $p$ = .008, $OR$ = 4.10). The main effect of embodiment was not significant ($b$ = 0.43, $SE$ = 0.50, $p$ = .384, $OR$ = 1.54). However, odds ratios revealed that in the incongruent condition, women were 54% more likely to choose the male candidate and men

**Table 8. Candidate choice by embodiment and participant gender in Study 2.**

| | Candidate Choice | | |
| --- | --- | --- | --- |
| | **Male** | **Female** | **$\chi 2, p$** |
| Congruent | | | 0.15, .696 |
| Male | 14 | 17 | |
| Female | 17 | 17 | |
| Incongruent | | | 7.44, .006** |
| Male | 9 | 24 | |
| Female | 20 | 13 | |
| Total | 60 | 71 | |

Note.

*$p < .05$

**$p < .01$.

were 167% more likely to choose the female candidate, whereas in the congruent condition, men were 21% more likely to choose the female candidate and women were equally likely to choose either candidate ($OR = 0$). Collectively, this suggests that incongruent embodiment exerted a substantial effect on candidate choice following perspective taking.

**Candidate ratings.** Differences in candidate ratings by experimental group are summarized in Table 9, while results of mixed ANOVA analyses are summarized in Table 10 and depicted in Fig 5. To meet the normality requirements of the mixed ANOVA, competence, hireability, likeability, and empathy were transformed using a log transformation with a reflection, as suitable for substantial negative skew, whereas IOS ratings showed substantial positive skew and were transformed using a log base 10 transformation. No significant effects were found for competence or hireability. Controlling for order effects, a significant main effect of gender was found for likeability, such that women rated candidates as more likeable than did men on average (moderate effect; Fig 5C).

We observed a significant two-way interaction between candidate gender and embodiment for IOS ratings, controlling for order effects (small to moderate effect; Fig 5E). Although participants reported greater interpersonal closeness with both candidates in the incongruent condition compared to the congruent condition, they rated the female candidate significantly higher in the incongruent condition and significantly lower in the congruent condition ($MD = 0.10$, $SE = 0.04$, $p = .015$). Significant two-way interactions with a moderate effect size also emerged between candidate gender and presentation order for IOS ($F_{(1, 113)} = 4.19$, $p = .043$, $\eta_p^2 = 0.04$) and likeability ratings ($F_{(1,126)} = 8.21$, $p = .005$, $\eta_p^2 = 0.06$).

**Table 9. Summary of candidate ratings by experimental group in Study 2.**

| Measure | Male Participants | | | | Female Participants | | | |
| --- | --- | --- | --- | --- | --- | --- | --- | --- |
| | Congruent | | Incongruent | | Congruent | | Incongruent | |
| | **Male** | **Female** | **Male** | **Female** | **Male** | **Female** | **Male** | **Female** |
| | *M (SD)* | | | | | | | |
| Competence | 8.203 (0.81) | 8.42 (0.94) | 8.32 (1.04) | 8.44 (1.18) | 8.50 (1.16) | 8.53 (1.05) | 8.61 (0.88) | 8.43 (1.16) |
| Hireability | 7.72 (1.03) | 7.70 (1.13) | 7.93 (1.43) | 7.99 (1.63) | 8.04 (1.34) | 8.25 (1.04) | 8.15 (1.11) | 7.86 (1.35) |
| Likeability | 6.65 (1.12) | 6.37 (1.44) | 6.70 (1.77) | 6.97 (1.71) | 7.11 (1.82) | 7.37 (1.69) | 7.37 (1.39) | 7.35 (1.47) |
| Empathy | 5.30 (1.65) | 5.11 (1.66) | 4.98 (1.77) | 5.38 (1.68) | 5.51 (1.94) | 5.92 (2.02) | 5.88 (1.92) | 5.80 (1.73) |
| IOS | 2.29 (1.01) | 2.17 (1.17) | 2.41 (1.10) | 2.63 (1.21) | 2.39 (1.36) | 2.34 (1.56) | 2.55 (1.18) | 2.73 (1.26) |

**Table 10. Mixed ANOVA models for candidate ratings following vr perspective taking.**

| Measure | Competence | Hireability | Likeability | Empathy | IOS |
|---|---|---|---|---|---|
| | $F_{(1, 114)}$ | $F_{(1, 114)}$ | $F_{(1, 126)}$ | $F_{(1, 114)}$ | $F_{(1, 113)}$ |
| Gender | 1.74 ($p$ = .190, $\eta_p^2$ = .01) | 1.56 ($p$ = .214, $\eta_p^2$ = .01) | 9.40** ($p$ = .003, $\eta_p^2$ = .07) | 6.02* ($p$ = .015, $\eta_p^2$ = .05) | 0.01 ($p$ = .926, $\eta_p^2$ < .01) |
| Embodiment | 0.21 ($p$ = .648, $\eta_p^2$ < .01) | 0.51 ($p$ = .478, $\eta_p^2$ < .01) | 1.00 ($p$ = .320, $\eta_p^2$ = .01) | 0.04 ($p$ = .839, $\eta_p^2$ < .01) | 3.36 ($p$ = .069, $\eta_p^2$ = .03) |
| Candidate | 1.10 ($p$ = .297, $\eta_p^2$ = .01) | 0.01 ($p$ = .908, $\eta_p^2$ < .01) | 0.49 ($p$ = .488, $\eta_p^2$ < .01) | 0.57 ($p$ = .452, $\eta_p^2$ < .01) | 0.03 ($p$ = .863, $\eta_p^2$ < .01) |
| Gender*Embodiment | 0.27 ($p$ = .604, $\eta_p^2$ < .01) | 2.53 ($p$ = .114, $\eta_p^2$ = .02) | 0.92 ($p$ = .339, $\eta_p^2$ = .01) | 0.14 ($p$ = .713, $\eta_p^2$ < .01) | 0.26 ($p$ = .611, $\eta_p^2$ < .01) |
| Candidate*Gender | 2.61 ($p$ = .109, $\eta_p^2$ = .02) | 0.48 ($p$ = .489, $\eta_p^2$ < .01) | 0.07 ($p$ = .786, $\eta_p^2$ < .01) | 0.08 ($p$ = .785, $\eta_p^2$ < .01) | 0.02 ($p$ = .897, $\eta_p^2$ < .01) |
| Candidate*Embodiment | 0.23 ($p$ = .635, $\eta_p^2$ < .01) | 0.08 ($p$ = .777, $\eta_p^2$ < .01) | 0.20 ($p$ = .654, $\eta_p^2$ < .01) | 0.02 ($p$ = .896, $\eta_p^2$ < .01) | 4.24* ($p$ = .042, $\eta_p^2$ = .04) |
| Candidate*Gender*Embodiment | 0.03 ($p$ = .866, $\eta_p^2$ < .01) | 1.71 ($p$ = .193, $\eta_p^2$ = .01) | 1.88 ($p$ = .173, $\eta_p^2$ = .02) | 4.81* ($p$ = .030, $\eta_p^2$ = .04) | 0.10 ($p$ = .760, $\eta_p^2$ < .01) |

*Note*. IOS = inclusion of the other in the self (i.e., self-other overlap); between-subjects factors: gender (female, male), embodiment (congruent, incongruent); within-subjects factor: candidate (female, male)

* $p$ < .05

** $p$ < .01.

Finally, a significant three-way interaction between candidate, gender, and embodiment (small to moderate effect; Fig 5D) suggested that men experienced greater empathy toward the female candidate when embodied in a female avatar and greater empathy toward the male candidate when embodied in a male avatar. Women exhibited a similar trend, reporting more empathy toward the male candidate when embodied in a male avatar and toward the female candidate when embodied in a female avatar.

**Mediation.** To test the hypothesis that virtual immersion exerts its influence on perspective taking via empathy, Model 4 from Hayes (2020) PROCESS macro version 3.5 was used to assess the pathways between embodiment and skills-based ratings for the female and male candidates (i.e., competence, hireability, likeability), with empathy and IOS ratings as mediators. Zero-order correlations are presented in Tables 11–13 summarises the indirect effects and effect sizes for each of the six models, with 0.02, 0.13, and 0.26 signifying small, moderate, and large effects, respectively [114, 115]. Fig 6 presents basic parallel mediation models assessing the hypothesized relationships between virtual immersion (x) and competence, hireability, and likeability (y) mediated by empathy (M1) and interpersonal closeness (M2) for the female and male candidates.

For the female candidate, virtual immersion significantly predicted empathy (i.e., A path), and independently, empathy significantly predicted hireability and likeability, but not competence ratings (i.e., B path). However, no significant mediating effects were observed. Virtual immersion did not predict interpersonal closeness, and interpersonal closeness did not predict any of the outcome ratings.

For the male candidate, the total indirect effect assessing the relationship between virtual immersion and competence ratings was significant, although neither empathy nor interpersonal closeness individually mediated this relationship. This suggests that only the combined influence of these factors contributed to mediation, and the effect size was small. For hireability and likeability ratings, empathy but not interpersonal closeness mediated the relationship between virtual immersion and these outcomes, and these effect sizes were moderate. Additionally, the total effect of virtual immersion on likeability was significant, whereas the direct

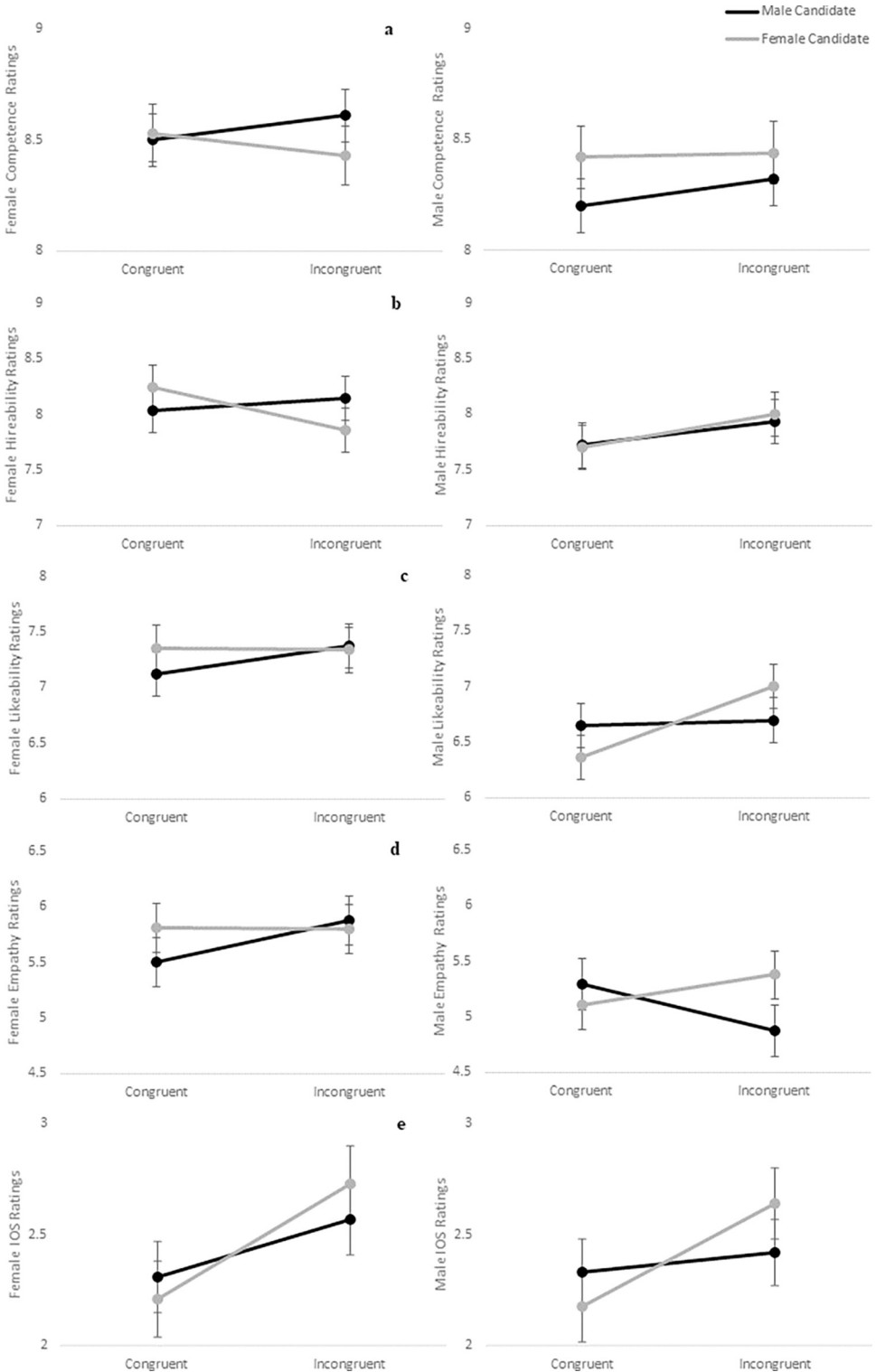

**Fig 5. Candidate ratings following vr perspective taking.** Comparison of female (left) and male (right) participant ratings by embodiment for the virtual candidates as illustrated for a) competence; b) hireability; c) likeability; d) empathy; e) interpersonal closeness.

**Table 11. Zero order correlations–female candidate.**

|  | 1 | 2 | 3 | 4 | 5 | 6 |
|---|---|---|---|---|---|---|
| 1. EQ Total | – |  |  |  |  |  |
| 2. Empathy | .19* | – |  |  |  |  |
| 3. IOS | .06 | .53** | – |  |  |  |
| 4. Competence | .01 | .04 | -.06 | – |  |  |
| 5. Hireability | -.04 | .22* | .12 | .80** | – |  |
| 6. Likeability | .10 | .67** | .42** | .33** | .53** | – |

effect was not, suggesting that immersion independently predicted likeability when holding constant empathy.

## Discussion

Incongruent embodiment substantially influenced candidate choice, particularly for men. Although men chose the female candidate significantly more often than women overall, the degree to which this occurred when embodied as a female avatar was significantly greater. Women exhibited a similar pattern, choosing the male candidate more often when embodied as male, compared to being equally likely to choose either candidate when embodied as female. Like Study 1, preference for the female candidate among men suggests a degree of socially desirable responding. Given the increasing visibility surrounding gender bias in STEM [116, 117], some men may consciously or unconsciously choose the female candidate to compensate for knowledge about undesirable behavior and conform to more favorable social expectations. However, socially desirable responding does not appear to be responsible for the impact of incongruent embodiment on men given the strength of the effect (*OR* = 2.67). Future work may seek to control for this factor and work to reduce tokenism in STEM (i.e., symbolically recruiting underrepresented individuals to meet diversity and inclusion metrics) [118] by implementing a standardized social desirability measure [119].

Further consistent with the online study, no significant differences in candidate ratings were found between experimental groups for competence, hireability, or likeability. However, the non-significant three-way trends in candidate ratings suggest that women tended to rate the male candidate higher and the female candidate lower when embodied as male, whereas men tended to rate the female candidate higher when embodied as female but also exhibited an increase in ratings of the male candidate. Ratings for empathy were consistent with candidate choice, with all participants reporting greater empathy toward the candidate who matched the gender of their assigned embodied avatar (Fig 5D), while interpersonal closeness with the female candidate significantly increased for all participants in the incongruent

**Table 12. Zero order correlations–male candidate.**

|  | 1 | 2 | 3 | 4 | 5 | 6 |
|---|---|---|---|---|---|---|
| 1. EQ Total | – |  |  |  |  |  |
| 2. Empathy | .24** | – |  |  |  |  |
| 3. IOS | .14 | .53** | – |  |  |  |
| 4. Competence | .03 | .26** | .21* | – |  |  |
| 5. Hireability | .02 | .43** | .27** | .80** | – |  |
| 6. Likeability | .16 | .67** | .38** | .48** | .64** | – |

*Note*. EQ = embodiment questionnaire, IOS = inclusion of other in the self scale.

Table 13. Indirect effects of immersion on competence, hireability, and likeability.

| | Effect (*SE*) | 95% CI | Effect Size (*SE*) | 95% CI |
|---|---|---|---|---|
| Competence (F) | | | | |
| Total | 0.02 (0.04) | -0.07, 0.10 | 0.01 (0.02) | -0.04, 0.06 |
| Empathy | 0.03 (0.04) | -0.06, 0.11 | 0.02 (0.03) | -0.03, 0.07 |
| IOS | -0.01 (0.02) | -0.07, 0.03 | -0.01 (0.01) | -0.04, 0.02 |
| Hireability (F) | | | | |
| Total | 0.08 (0.05) | -0.02, 0.19 | 0.04 (0.03) | -0.01, 0.10 |
| Empathy | 0.08 (0.05) | -0.02, 0.20 | 0.04, (0.03) | -0.01, 0.10 |
| IOS | 0.01 (0.02) | -0.04, 0.05 | 0.01 (0.02) | -0.02, 0.02 |
| Likeability (F) | | | | |
| Total | 0.30 (0.17) | -0.02, 0.64 | 0.12 (0.06) | -0.01, 0.24 |
| Empathy | 0.28 (0.16) | -0.02, 0.63 | 0.11 (0.06) | -0.01, 0.24 |
| IOS | 0.01 (0.02) | -0.03, 0.07 | 0.01 (0.01) | -0.01, 0.03 |
| | Effect (*SE*) | 95% CI | Effect Size (*SE*) | 95% CI |
| Competence (M) | | | | |
| Total | 0.07 (0.04) | 0.01, 0.16* | 0.04 (0.03) | 0.01, 0.11* |
| Empathy | 0.04 (0.04) | -0.02, 0.14 | 0.03 (0.03) | -0.02, 0.09 |
| IOS | 0.03 (0.03) | -0.01, 0.09 | 0.02 (0.02) | -0.01, 0.06 |
| Hireability (M) | | | | |
| Total | 0.17 (0.07) | 0.05, 0.31* | 0.10, (0.04) | 0.03, 0.27* |
| Empathy | 0.15 (0.06) | 0.04, 0.29* | 0.13 (0.06) | 0.04, 0.25* |
| IOS | 0.02 (0.03) | -0.03, 0.10 | 0.02 (0.02) | -0.02, 0.06 |
| Likeability (M) | | | | |
| Total | 0.34 (0.14) | 0.11, 0.64* | 0.15 (0.07) | 0.05, 0.27* |
| Empathy | 0.33 (0.14) | 0.10, 0.63* | 0.15 (0.07) | 0.05, 0.27* |
| IOS | 0.01 (0.03) | -0.03, 0.08 | 0.01 (0.01) | -0.02, 0.04 |

*Note*. Completely standardized effects are reported as a measure of effect size, F = female candidate, M = male candidate, CI = confidence interval.

condition (Fig 5E). In contrast to Study 1, virtual reality perspective taking appears to have some influence on both men's and women's perceptions of a candidate based on gender presentation.

Although women reported greater likeability of either candidate than did men on average, order effects revealed an unexpected pattern for IOS and likeability ratings. While participants expressed greater interpersonal closeness toward the candidate presented second, they also expressed greater likeability toward the candidate presented first. These results could simply be an artifact of the current methodology if, for instance, evaluations of the candidate presented first were skewed by distractions associated with becoming accustomed to the virtual experience. However, order of candidate presentation could also have an impact in the real world, and thus, these effects require investigation in future work.

Finally, mediation analyses mirrored regression analyses from Study 1. That is, empathy but not interpersonal closeness mediated the relationship between virtual immersion and candidate ratings, but only for the male candidate. For the female candidate, virtual immersion was associated with empathy, but not interpersonal closeness, and empathy was associated with hireability and likeability, but not competence. However, no mediating effects were found. Consistent with online perspective taking, objective evaluations of candidate skill level are maintained for a male target, but not for a female target following virtual reality perspective

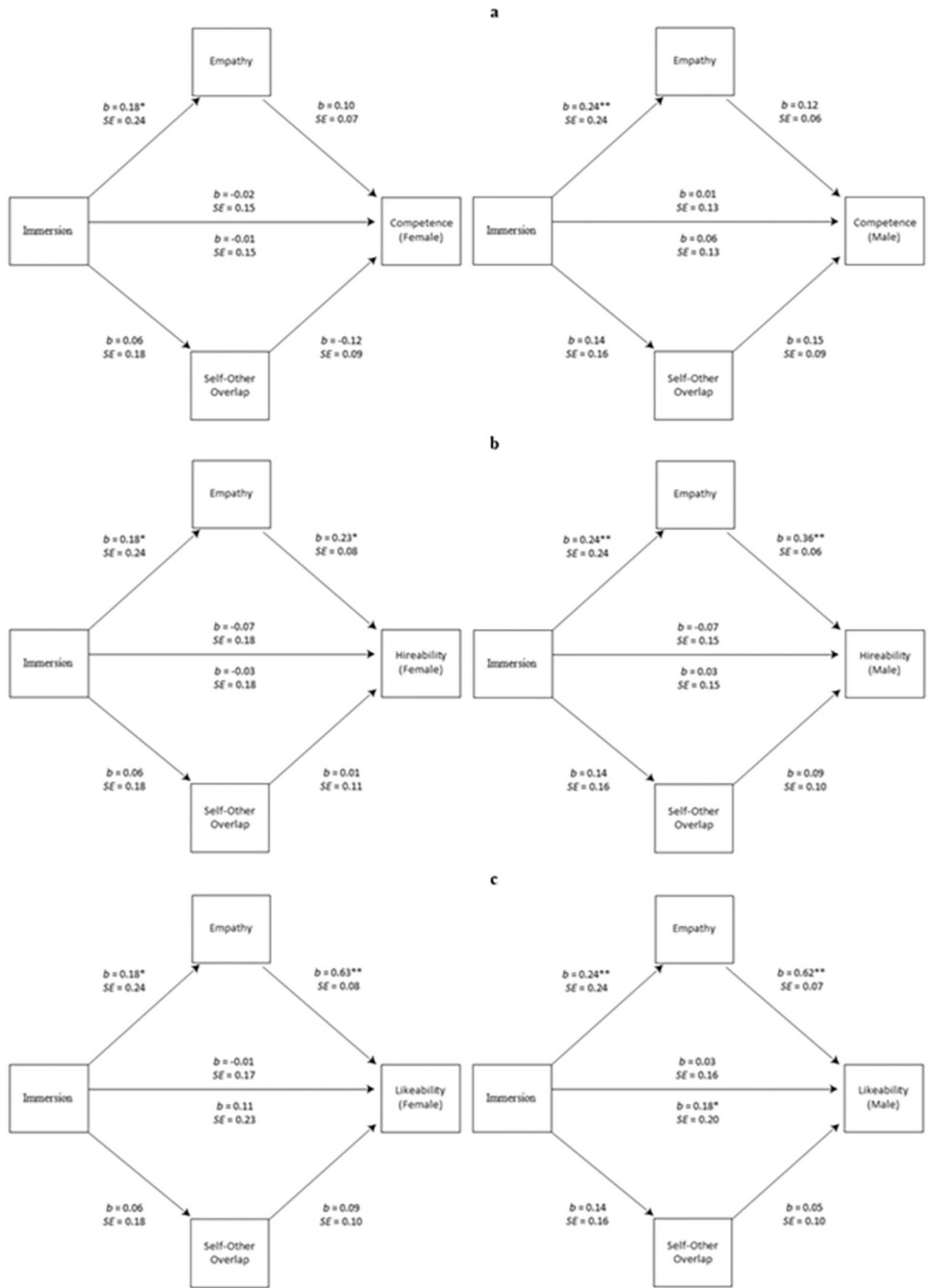

**Fig 6. Mediations.** a) competence; b) hireability; c) likeability. Total effect is presented below the horizontal line, while the direct effect of X on Y is presented above.

taking and observed increases in empathy do not appear to transfer to more equitable perceptions of candidate skill level, particularly for competence.

## General discussion

Previous research has applied virtual perspective taking techniques to modulate unconscious bias, empathy, and interpersonal behavior toward marginalized individuals [79]. Accordingly,

we investigated whether such an approach could be applied to binary gender. Using an online platform (Study 1) and fully immersive virtual reality (Study 2), we evaluated the impact of congruent and incongruent virtual embodiment on the rating and selection of a female and male candidate in the context of a STEM interview. Although between-group differences were not established following virtual self-representation in the online platform, the experience of fully immersive embodiment substantially diverged for men and women following exposure to virtual reality perspective taking. Empathy emerged as a meaningful contributor to variance in candidate ratings, whereas interpersonal closeness did not, suggesting the process of behavioral change following virtual embodiment may initially depend on changes in empathy.

Although our hypotheses regarding candidate choice (H1, H5) were not statistically supported in Study 1, results of Study 2 offer support for the assumption that individuals take on the identity of their assigned avatar and subsequently exhibit greater identification with co-actors of similar identities. That is, men embodied as female chose the female candidate more often (supporting H1), and women chose the male candidate more often when embodied as male (supporting H5). Moreover, the effect incongruent embodiment on candidate choice was more pronounced for men, in support of H3.

Although our hypotheses for candidate ratings (H2, H5) were not statistically supported by either study, a lack of statistical significance and small effect sizes ($\eta_p^2 \leq .03$) are not overly surprising given that candidates were equivalent in their skill level and qualifications. Nevertheless, candidate rating trends were consistent with outcomes for candidate choice, suggesting that seemingly inconsiderable changes in candidate evaluations could indeed inform subsequent and material differences in decision making.

Empathy ratings were also consistent with candidate choice outcomes. Notably, the preliminary effect of empathy established in Study 1 (for men) was replicated and extended by the larger sample in Study 2 (for both women and men), with embodiment and gender exerting a moderate effect on variation in empathy, but not interpersonal closeness, toward the candidates (in partial support of H4). However, women consistently reported greater empathy than men. Thus, it may be the case that virtual embodiment is more successful in modulating gender bias for men than for women due to experiences of empathy. That is, women may benefit from increases in empathy to a lesser degree if they are already substantially more empathetic than men.

Consistent with extensive research on the Proteus effect [12], multiple regression (Study 1) and mediation analyses (Study 2) support the idea that empathy might underlie the relationship between virtual immersion and behavioral change. However, the present study is the first, to our knowledge, to statistically evaluate the linear relationships between immersion, empathy, and behavior. If empathy is indeed the mechanism by which virtual immersion can lead to change, researchers could take a more nuanced approach to modulating behaviors stemming from unconscious biases and explicitly target empathy. Rather than simply using virtual embodiment to evaluate how individuals behave when they become a character or persona in virtual reality (e.g., doctor, villain, celebrity), embodied perspective taking may attenuate society-level stereotypes about marginalized identities if applied authentically to contexts in which unconscious biases arise.

## Theoretical implications

Taken together, our results suggest that virtual perspective taking may be most successful when undertaken in a fully immersive virtual environment. Although online perspective taking exerted some influence on changing baseline perceptions about female and male candidates for a STEM position, this influence was minimal, restricted to men, and did not have an

impact on tendencies toward in-group bias. The substantial differences between online and virtual reality perspective taking can be attributed to the experience of embodiment and the way in which the perceptual sense of body ownership over the virtual avatar can operate across these two modalities.

Online perspective taking lacks the immersive components of fully interactive virtual reality and thus, can only offer an analogue to true embodiment through virtual self-representation on a two-dimensional screen, where viewpoint alone is controlled using a mouse or trackpad. In comparison, immersive virtual reality offers more socially relevant, nonverbal, and perceptual cues, which comprise the perceptual sense of body ownership [11, 96, 111, 120]. To understand how decision making is informed by underlying cognitive biases and beliefs about gender stereotypes, perceptual and motor systems must be engaged during the interactions one undertakes within their environment [121–123]. Body ownership, motor control, and agency are critical to the perceptual and motor sensations of embodiment in the real-world [124, 125], but these components are substantially reduced in an online format. Therefore, virtual reality perspective taking provides us with a more valid approach to experimentally evaluating embodiment and subsequent behavioral modulation.

Research has shown that both demographic and social familiarity, affiliation, and similarity contribute to increased empathy toward perceived in-group members [126, 127]. Moreover, group affiliations are malleable and can be influenced by self-perception, emotion regulation, and self-regulation processes [96]. Specifically, virtual embodiment can allow for bodily and conceptual changes to self-perception, which increase in-group affiliation, decrease negative out-group evaluation, and in turn, increase empathy [95]. In the present study, such modulations to self-perception (i.e., gender-incongruent embodiment) corresponded to changes in empathy consistent with malleable in-group affiliation (e.g., men embodied in a female avatar come to perceive a female target as part of the temporary in-group). In turn, empathy corresponded to behavioral changes that indicate acute modulation of gender bias (i.e., candidate choice, candidate ratings), and this effect appeared to be stronger for men than for women. However, future research should seek to clarify whether acute behavioral changes can be maintained and whether the influence of interpersonal closeness may come about in the long term.

Notably, perspective taking interventions targeting empathy alone are unlikely to be sufficient in addressing the underrepresentation of women in STEM. Although we did not find significant differences between groups for objective candidate ratings (i.e., competence, hireability, likeability), rating trends for these objective factors were consistent with aforementioned subjective feelings of empathy. Taken together, small differences in objective measurements and substantial differences in subjective perceptions can be consequential for decision making, in particular under conditions of uncertainty or ambiguity [128, 129]. Moreover, subjective intentions do not always correlate with behavioral outcomes [130, 131]. Consistent with previous research, empathy did not predict competence ratings for the female candidate in either study. Because women are consistently rated less competent than otherwise equivalent men, particularly in STEM positions [60, 63], industry strategies must target this factor more explicitly and expedite interventions for its refinement. To improve unconscious gender bias at the level of the STEM interview, procedures should ensure not only that candidate ratings are predetermined using transparent and objective criteria, but also that contextually relevant perspective taking is implemented before undertaking critical decisions that remain vulnerable to bias.

Although fully immersive perspective taking in virtual reality may not yet be practical for STEM institutions, the present study contributes to the emerging necessity of ecologically relevant contextual exposure during perspective taking for bias [86, 87]. Thus, future work in this area should seek to reduce reliance on the Gender-Career IAT and embodiment procedures

undertaken with limited contexts [84, 85] and instead, begin to integrate more pragmatic virtual interactions, including those with live co-actors. Such an approach may not only allow the research community to better apply recommendations to industry or institutional diversity and inclusion operations, but also reduce the impact of previous mixed findings, which work in opposition to equity in STEM when used as foundations for arguments of biological essentialism [132] or reverse gender bias [133].

## Strengths, limitations, and future research

Quantitative results and derived theoretical implications should be considered with respect to the following limitations. First, as all avatars were designed neutral in appearance, we could not account for intersectionality. Experiences of discrimination and bias are compounded for those who are marginalized by more than one category of identity, for instance women of color in STEM [63, 134], and future research should seek to extend embodiment assessments to contemporaneous identity categories (e.g., race, gender). Second, we employed a binary view of gender. However, modern sexism also encompasses the belief that cisgender individuals (i.e., those who identify with their assigned gender at birth) are superior to those who are gender diverse [6, 135]. Future investigations should integrate androgynous, gender ambiguous, transgender, and non-binary virtual avatars to further assess the influence of embodiment on gender bias toward gender-diverse individuals. Finally, we did not measure degree of immersion in the online study. However, a standardized quantitative comparison assessing differences in this factor among traditional, online, and virtual reality perspective taking approaches is warranted, as only one study to date has compared differences in perspective taking for bias across these modalities [15]. Although the Gonzalez-Franco and Peck [111] measure is intended for use in virtual reality, future research may benefit from its inclusion to clarify degree of immersion in comparative assessments.

Despite these limitations, this research offers novel contributions to the existing embodiment literature and extends the methodologies of previous hiring studies by using virtual environments that allow for the inclusion of visual and auditory components that signify gender, which are rarely blinded during interpersonal tasks. Furthermore, this study extends the emerging research on understanding bias and discrimination through virtual embodiment by comparing the effects of congruent and incongruent embodiment across genders. While initial work suggests that virtual perspective taking reduces bias and increases prosocial behavior compared to non-embodied perspective taking [13, 15], few studies have used sufficient control groups to investigate whether the effects of embodiment differed across members of the populations of interest [87].

In particular, our findings for empathy and interpersonal closeness provide a compelling case for future embodiment research to include measures of empathy with greater dimensionality. While cognitive empathy is traditionally related to perspective taking, affective empathy may be more closely related to feelings interpersonal closeness [136]. Given that interpersonal closeness may not arise as a contributing factor until several weeks post-immersion [15], an understanding of how empathy operates in this context may offer clarity on the roles of cognition and emotion for unconscious bias. Moreover, we can now begin to explore these phenomena alongside measurable nonverbal behaviors such as postural sway, behavioral mimicry, and eye-tracking, which may differ depending on one's implicit and explicit biases [137]. Such multifaceted assessments may provide greater detail in differentiating behavioral outcomes, like a small effect for candidate ratings alongside a large effect for candidate choice. If we can begin to identify the systems underlying vulnerability to social biases in decision making, we can effect change on these systems to improve social equity.

## Conclusion

By employing virtual perspective taking during a simulated STEM interview across online and virtual reality platforms, we discover that the experience of embodiment for binary gender can allow both men and women to experience first-hand what it might be like to be in the body of the other. Greater immersion led to greater identification with an incongruent gender identity, and empathy may be a key mechanism which underlies this phenomenon. Although temporary, the experience of being immersed in the environment and in the body of the avatar allows individuals to more deeply engage with the perspective taking experience through contextually relevant interpersonal interaction. Extending approaches from previous hiring studies, we offer the STEM community a timely suggestion to improve the underrepresentation of women and gender diverse individuals by targeting intervention and training at more equitable characterizations of competence. We highlight pivotal avenues for future research regarding gender and bias in hiring, in which greater dimensionality in measures of empathy and nonverbal behavior will be key to understanding how social biases shape decision making, in particular when applied to gender diverse representation.

While the future of virtual perspective taking as an intervention requires further inquiry, it is anticipated that it could change attitudes and behaviors in unforeseen ways. Where women remain underrepresented in male-dominated industries, they also remain underrepresented in decision-making processes that disproportionately disadvantage their progress and devalue their contributions. If we can enhance the ability of those individuals who benefit from systemic privilege to not only imagine, but also experience first-hand what it might be like to be someone else, we might better address gender representation in STEM as well as the underrepresentation of marginalized individuals across other domains.

## Supporting information

**S1 Data.**
(DOCX)

## Acknowledgments

The authors would like to thank Michael J. Richardson, Meredith Porte, and Gaurav Patil for their helpful comments on the preliminary manuscript and for their support in creating the virtual environment and virtual avatars used in this research.

## Author Contributions

**Conceptualization:** Cassandra L. Crone, Rachel W. Kallen.

**Data curation:** Cassandra L. Crone.

**Formal analysis:** Cassandra L. Crone.

**Investigation:** Cassandra L. Crone.

**Methodology:** Cassandra L. Crone, Rachel W. Kallen.

**Supervision:** Rachel W. Kallen.

**Visualization:** Cassandra L. Crone.

**Writing – original draft:** Cassandra L. Crone.

**Writing – review & editing:** Cassandra L. Crone, Rachel W. Kallen.

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
