## [Decision Letter · Decision Letter 0]

7 Apr 2022

PONE-D-21-34549Interview with an Avatar: Comparing Online and Virtual Reality Perspective Taking for Gender Bias in STEM Hiring DecisionsPLOS ONE

Dear Dr. Crone,

Thank you for submitting your manuscript to PLOS ONE. After careful consideration, we feel that it has merit but does not fully meet PLOS ONE’s publication criteria as it currently stands. Therefore, we invite you to submit a revised version of the manuscript that addresses the points raised during the review process.

First of all, I want to congratulate the authors for carrying out such a robust study.

This manuscript is convincingly ambitious, sound, credible, and has soundness and credible methodology.

- The topic of this manuscript is interesting, and it introduces an innovative aspect.  Theme is relevant and has a conceptual domain, the study is well conceived, well crafted, and well-presented paper.

- The methods are appropriate, accurate, and objectivity for the experiments, and improves the understanding of the reader.

- The analyses are while not complex, informative, and appropriate for the experiments.

- The authors recognize the limitations of this research, and in my view, this manuscript has main value and provides evidence that the authors seem to recognize the main innovation of their study,

- Overall, the paper is well placed to stimulate future research within the field.

- It is clear a substantial amount of work has gone into preparing this manuscript, and it can be reconsidered able to be published and may result in a modest contribution to the literature, after major revision:

References /Citations shown little inconsistencies / unconformities: Please revise.

We look forward to receiving your revised manuscript.

Kind regards,

Sonia Brito-Costa, Ph.D.

Academic Editor

PLOS ONE

Journal Requirements:

2. Please note that according to our submission guidelines (http://journals.plos.org/plosone/s/submission-guidelines), outmoded terms and potentially stigmatizing labels should be changed to more current, acceptable terminology. To this effect,  “Caucasian” should be changed to “white” or “of [Western] European descent” (as appropriate).

Reviewers' comments:

Reviewer's Responses to Questions

**Comments to the Author**

1. Is the manuscript technically sound, and do the data support the conclusions?

Reviewer #1: Yes

2. Has the statistical analysis been performed appropriately and rigorously? 

Reviewer #1: Yes

3. Have the authors made all data underlying the findings in their manuscript fully available?

Reviewer #1: Yes

4. Is the manuscript presented in an intelligible fashion and written in standard English?

Reviewer #1: Yes

5. Review Comments to the Author

Reviewer #1: The core content presented in this manuscript is interesting, namely to examine the role of virtual perspective taking for binary gender using an online platform and immersive virtual reality. The information provided allows forming an overview on the proposed research. The article is overall well structured and fairly easy to read. The methodology is good and the analysis is convincing.

Regards

6. PLOS authors have the option to publish the peer review history of their article (what does this mean?). If published, this will include your full peer review and any attached files.

Reviewer #1: No

---

## [Author Response · Author response to Decision Letter 0]

5 May 2022

The authors thank the editor and reviewer for taking the time to evaluate our research and for providing such positive comments in their reviews. Below, we provide our responses to the suggested revisions. 

Journal Requirements:

Comment 1. Please ensure that your manuscript meets PLOS ONE's style requirements, including those for file naming. The PLOS ONE style templates can be found at https://journals.plos.org/plosone/s/file?id=wjVg/PLOSOne_formatting_sample_main_body.pdf and https://journals.plos.org/plosone/s/file?id=ba62/PLOSOne_formatting_sample_title_authors_affiliations.pdf

Response 1. The authors have revised the formatting of the manuscript to be consistent with PLOS ONE’s style requirements as linked above. Specifically, levels of heading, line numbering, and title page requirements have been adjusted accordingly. File naming for all additional files have been adjusted to match the journal conventions. Finally, all tables directly follow the paragraph in which they are first mentioned within the text. Formatting and style of citations and references are addressed below in comments 4 and 5. 

Comment 2. Please note that according to our submission guidelines (http://journals.plos.org/plosone/s/submission-guidelines), outmoded terms and potentially stigmatizing labels should be changed to more current, acceptable terminology. To this effect, “Caucasian” should be changed to “white” or “of [Western] European descent” (as appropriate).

Response 2. Instances of the word “Caucasian” have been adjusted to “White”, specifically in Tables 1 and 6.

Comment 3. We note that you have stated that you will provide repository information for your data at acceptance. Should your manuscript be accepted for publication, we will hold it until you provide the relevant accession numbers or DOIs necessary to access your data. If you wish to make changes to your Data Availability statement, please describe these changes in your cover letter and we will update your Data Availability statement to reflect the information you provide.

Response 3. The DOI associated with the published dataset at Mendeley Data has now been provided.

Comment 4. Please review your reference list to ensure that it is complete and correct. If you have cited papers that have been retracted, please include the rationale for doing so in the manuscript text, or remove these references and replace them with relevant current references. Any changes to the reference list should be mentioned in the rebuttal letter that accompanies your revised manuscript. If you need to cite a retracted article, indicate the article’s retracted status in the References list and also include a citation and full reference for the retraction notice.

Response 4. The authors have reviewed the references to ensure that all citations within the body of the manuscript are also included in the reference list and that there are no additional references included without citation in the paper. Cited material has been checked for editor concern via the online platform scite. As the Lindsey et al. (2015) article (reference 73) was flagged for editorial notices, we have now cited the published Erratum associated with this article (reference 75) alongside any instances of the original article, specifically on lines 122 and 126. The correction published in the Erratum did not affect the original authors’ claims or interpretations, as it was only concerned with the second row of data presentation in Table 2 of the original article. 

Comment 5. References /Citations shown little inconsistencies / unconformities: Please revise.

Response 5. The references and citations have been revised to ensure consistency across reference type. Additionally, URLs have been added for online articles and reports. References and citations have further been examined for any formatting errors, which have now been corrected. Finally, we have updated the references and citations to match the reference style of the journal, using EndNote software. Citations are numbered within brackets in order of appearance within the body of the manuscript and appear in this order within the reference list.

Response to Reviewer #1: The authors thank Reviewer #1 for their efforts in reviewing the manuscript and for providing their comments as copied below. We note that the reviewer did not request any specific revisions, and thus, we have simply included their review here for reference. 

Reviewers' comments:

Reviewer's Responses to Questions

Comments to the Author

1. Is the manuscript technically sound, and do the data support the conclusions?

Reviewer #1: Yes

2. Has the statistical analysis been performed appropriately and rigorously?

Reviewer #1: Yes

3. Have the authors made all data underlying the findings in their manuscript fully available?

Reviewer #1: Yes

4. Is the manuscript presented in an intelligible fashion and written in standard English?

Reviewer #1: Yes

5. Review Comments to the Author

Reviewer #1: The core content presented in this manuscript is interesting, namely to examine the role of virtual perspective taking for binary gender using an online platform and immersive virtual reality. The information provided allows forming an overview on the proposed research. The article is overall well structured and fairly easy to read. The methodology is good and the analysis is convincing.

Regards

6. PLOS authors have the option to publish the peer review history of their article (what does this mean?). If published, this will include your full peer review and any attached files.

Do you want your identity to be public for this peer review? For information about this choice, including consent withdrawal, please see our Privacy Policy.

Reviewer #1: No

---

## [Editor Report · Decision Letter 1]

23 May 2022

Interview with an Avatar: Comparing Online and Virtual Reality Perspective Taking for Gender Bias in STEM Hiring Decisions

PONE-D-21-34549R1

Dear Dr. Crone,

We’re pleased to inform you that your manuscript has been judged scientifically suitable for publication and will be formally accepted for publication once it meets all outstanding technical requirements.

Kind regards,

Sónia Brito-Costa, Ph.D.

Academic Editor

PLOS ONE
---

## [Editor Report · Acceptance letter]

25 May 2022

PONE-D-21-34549R1 

Interview with an Avatar: Comparing Online and Virtual Reality Perspective Taking for Gender Bias in STEM Hiring Decisions 

Dear Dr. Crone:

I'm pleased to inform you that your manuscript has been deemed suitable for publication in PLOS ONE. Congratulations! Your manuscript is now with our production department. 

Kind regards, 

on behalf of

Dr. Sónia Brito-Costa 

Academic Editor

PLOS ONE